# Discovery of quantum phases in the Shastry-Sutherland compound SrCu$_2$(BO$_3$)$_2$ under extreme conditions of field and pressure

Zhenzhong Shi [1,8], Sachith Dissanayake[1], Philippe Corboz[2], William Steinhardt [1], David Graf [3], D. M. Silevitch [4], Hanna A. Dabkowska[5], T. F. Rosenbaum[4], Frédéric Mila [6] & Sara Haravifard [1,7✉]

The 2-dimensional layered oxide material SrCu$_2$(BO$_3$)$_2$, long studied as a realization of the Shastry-Sutherland spin topology, exhibits a range of intriguing physics as a function of both hydrostatic pressure and magnetic field, with a still debated intermediate plaquette phase appearing at approximately 20 kbar and a possible deconfined critical point at higher pressure. Here, we employ a tunnel diode oscillator (TDO) technique to probe the behavior in the combined extreme conditions of high pressure, high magnetic field, and low temperature. We reveal an extensive phase space consisting of multiple magnetic analogs of the elusive supersolid phase and a magnetization plateau. In particular, a 10 × 2 supersolid and a 1/5 plateau, identified by infinite Projected Entangled Pair States (iPEPS) calculations, are found to rely on the presence of both magnetic and non-magnetic particles in the sea of dimer singlets. These states are best understood as descendants of the full-plaquette phase, the leading candidate for the intermediate phase of SrCu$_2$(BO$_3$)$_2$.

[1] Department of Physics, Duke University, Durham, NC 27708, USA. [2] Institute for Theoretical Physics and Delta Institute for Theoretical Physics, University of Amsterdam, Science Park 904, 1098 XH Amsterdam, The Netherlands. [3] National High Magnetic Field Laboratory, Florida State University, Tallahassee, FL 32310, USA. [4] Division of Physics, Math and Astronomy, California Institute of Technology, Pasadena, CA 91125, USA. [5] Brockhouse Institute for Material Research, McMaster University, Hamilton, ON L8S 4M1, Canada. [6] Institute of Physics, Ecole Polytechnique Fédérale de Lausanne (EPFL), CH-1015 Lausanne, Switzerland. [7] Department of Mechanical Engineering and Materials Science, Duke University, Durham, NC 27708, USA. [8] Present address: Institute for Advanced Study, School of Physical Science and Technology, Soochow University, Suzhou 215006, China. ✉email: sara.haravifard@duke.edu

While the behavior of individual spins in isolation is well understood, complex behavior and exotic quantum states often emerge from networks of such spins, especially when competing interactions forestall the formation of simple ordered states, a phenomenon known as magnetic frustration[1]. A key tool for understanding these states is the ability to tune parameters such as the relative strength of the different interactions or the external magnetic field. In that respect, the Shastry–Sutherland (SS) model[2], a 2-dimensional (2D) network of orthogonal interacting spin dimers, together with its experimental realization $SrCu_2(BO_3)_2$, are prominent candidates.

The Cu spins 1/2 in $SrCu_2(BO_3)_2$ form weakly coupled 2D networks of orthogonal dimers topologically equivalent to the SS model. For the pure Heisenberg model, the exact ground state is a product of singlet dimers[2,3] as long as the inter-dimer coupling $J'$ is not too large as compared to the intra-dimer coupling $J$. Heuristically, we can think of the magnetization as due to magnetic particles $T_1$ that form when a dimer singlet $S$ is replaced by a triplet polarized along the field. These particles have a very small kinetic energy due to the highly frustrated dimer arrangement, leading to very flat bands and to Mott insulating phases (i.e., magnetization plateaus) at fractional fillings. The first magnetization plateaus[4] in $SrCu_2(BO_3)_2$, at 1/8 and 1/4, were initially observed in 1999, and the confirmation that the translational symmetry is broken in the 1/8 plateau soon followed[5]. At ambient pressure, additional plateaus have been identified[6–13] to build the improbable sequence 1/8, 2/15, 1/6, 1/4, 1/3, 2/5, and 1/2. It required 15 years and the invention of tensor network algorithms to develop a theory capable of accounting for this remarkable series[14]. Some of these plateaus (1/4, 1/3, 1/2) can be simply interpreted as Wigner crystals of $T_1$ particles, while the lower magnetization plateaus are best seen as Wigner crystals of spin-2 bound states that form because of a second-order kinetic term in $J'/J$ that leads to a binding between pairs of $T_1$ particles on neighboring parallel dimers[14]. Additionally, the supersolid phases correspond to adding $T_1$ particles to a plateau phase, the hopping of these extra-particles being due to correlated hopping that takes advantage of the underlying network of $T_1$ particles[15,16]. Between the plateaus, translation invariance is never restored, and it remains a challenge to establish which of these intermediate phases are spin-supersolids and which are incommensurate phases with proliferating domain walls[11]. The excitation spectrum is also remarkable, with very flat bands, and it has been shown that, due to Dzyaloshinskii–Moriya interactions[17–21], a small field induces topological magnon bands with non-zero Chern numbers[22] and experimental consequences still to be explored.

In addition to the rich set of physics revealed by high magnetic fields, $SrCu_2(BO_3)_2$ is remarkably sensitive to pressure for an oxide, and two phase transitions have been observed in it[23–29]. This sensitivity is enabled by the geometry of the Cu-Cu bonds: at ambient pressure, the intra-dimer Cu–Cu bond is close to 90°, and applying pressure brings this angle even closer to 90°, reducing $J$ and increasing the ratio $J'/J$ (ref. [29,30]). At ambient pressure, the ratio $J'/J \simeq 0.63$ for $SrCu_2(BO_3)_2$[12] puts it close to the boundary of the dimer phase, hence a relatively modest pressure of order 20 kbar is sufficient to induce a first-order transition into another gapped phase[23–27], followed at higher pressure by a transition into another phase still to be fully characterized[27,31]. Similarly, the phase diagram of the SS model has three phases[32–36]: an exact dimer phase up to $J'/J \simeq 0.675$, an antiferromagnetic (AFM) phase above $J'/J = 0.765(15)$(ref. [36]) (in the limit $J'/J \longrightarrow \infty$ the SS model is equivalent to the square lattice antiferromagnet), and an intermediate plaquette phase in between, where strong $J'$ bonds form around half the empty square plaquettes of the SS lattice.

The transition between the dimer phase and the intermediate phase is clearly first order, and it has been shown very recently that as a function of temperature it terminates at a critical point analogous to that of water[31]. By contrast, the nature of the transition between the intermediate phase and the AFM phase is still debated, and the interest in this transition has risen recently after the proposal that it could be a deconfined quantum critical point[37–39]. NMR experiments have revealed early on that there are two types of Cu sites[23], inconsistent with the intermediate phase of the SS model, and various experimental results seem to be rather consistent with a full-plaquette phase where strong $J'$ bonds form around half the square plaquettes that contain a dimer. The emerging picture for $SrCu_2(BO_3)_2$ is then that of a system dominated by a tendency to an orthorhombic distortion at intermediate pressure[23,25,28]. In the absence of direct probes of the symmetry of this intermediate phase, its precise nature remains an open issue, and a very important one because the nature of the intermediate phase will of course influence the nature of the transition into the AFM phase.

In this paper, our aim is to gain insight into the properties of $SrCu_2(BO_3)_2$ by studying its high-field properties in the relevant pressure range using tunnel diode oscillator (TDO) technique, and by an investigation of the high-field properties of the SS model in the corresponding $J'/J$ range using tensor network methods. These regions of the phase diagrams of $SrCu_2(BO_3)_2$ and of the SS model have not been previously explored and here are demonstrated to host exotic magnetic phases, including a $10 \times 2$ supersolid and a 1/5 plateau. Furthermore, we show that the discovered complex magnetic phase diagram of $SrCu_2(BO_3)_2$, at high-pressure and high-magnetic field, offers insight into the second phase transition, suggesting it to display a possible deconfined quantum critical point with fractional excitations. Our results set the ground for further studies of $SrCu_2(BO_3)_2$ under pressure. We, additionally, establish TDO as a viable and effective technique to be utilized for similar measurements for other quantum magnets under combined extreme conditions of high $H$, high $P$, and low $T$.

## Results

**Experimental results.** The TDO technique has been previously proven to be a valuable tool[13,40,41] for probing the behavior of pure $SrCu_2(BO_3)_2$ in the spin-dimer phase and the ambient-pressure behavior of doped $SrCu_{2-x}Mg_x(BO_3)_2$. It allows measurement of the change in magnetization at sub-Kelvin temperatures, high pressures, and high magnetic fields (see "Methods" for details), making it especially well-suited for physical systems that require all three simultaneously. In Fig. 1a, we show TDO magnetic susceptibility measurements for pure $SrCu_2(BO_3)_2$ in $\mu_0 H$ up to 45 T ($H \| ab$) and $T = 0.3$ K, where $df/dH \propto dM^2/d^2H$ (ref. [40]), for a series of pressures spanning the spin-dimer and the putative 4-spin plaquette phases. The high sensitivity of the technique allows the identification of weak magnetization changes that would otherwise be extremely difficult to detect. For example, at $P = 0$, we identify seven anomalies in $M(H)$ at fields $H_1 \sim 27.5$ T, $H_2 \sim 30.2$ T, $H_3 \sim 31.8$ T, $H_4 \sim 33.4$ T, $H_5 \sim 34.4$ T, $H_6 \sim 37.6$ T, and $H_7 \sim 43.6$ T, all of which correspond to jumps or slope changes in magnetization (see "Methods" for details).

We first focus on the two anomalies at the highest fields at $P = 0$, namely $H_6$ and $H_7$ (Fig. 1a), which can be identified immediately as the onset of the 1/4 and 1/3 magnetization plateaus, respectively[40]. The natural next step is to follow the two anomalies to higher pressures. At $P = 1.1$ GPa, two similar anomalies are also observed, though shifted to lower fields ($\sim 35$ T and $\sim 40$ T, respectively). In the intermediate plaquette

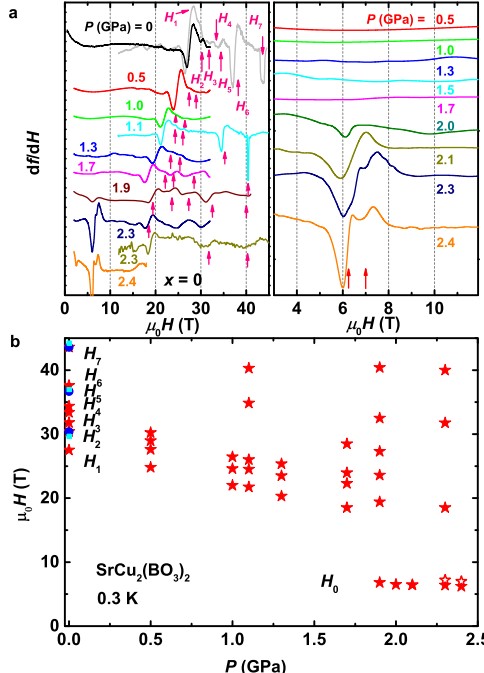

**Fig. 1 $P$-dependence of the magnetization plateaus and emergence of low-field anomalies in $SrCu_2(BO_3)_2$. a** (left panel): $df/dH$ vs. $H$ for $P$ up to 2.4 GPa at 0.3 K. The data consist of results from multiple runs on different samples using a 18 T superconducting magnet, a 35 T resistive magnet, and a 45 T hybrid magnet ($H\|ab$ for all measurements). Red arrows denote $H_1$, $H_2$, $H_3$, $H_4$, $H_5$, $H_6$, and $H_7$ at $P = 0$. $H_1$, $H_2$, $H_6$, and $H_7$ correspond to the sub-1/8 anomaly and the 1/8, 1/4, 1/3 plateaus, respectively. $H_3$, $H_4$, and $H_5$ are likely intermediate 2/15, 1/7, and 1/6 plateaus. The 1/8 plateau is identified as the shoulder that appears at a slightly higher $H$ than the large sub-1/8 anomaly (see Supplementary Fig. 1; for identification of the other features such as the high-$P$ 1/5 plateau, see "Methods" and Supplementary Fig. 2). **a** (right panel) Magnified view of the low-field behavior, showing the emergence of the low-field anomaly, which splits above $P \sim 2.2$ GPa as indicated by the two red arrows. The 2.3 GPa and 2.4 GPa traces are from measurements on two different samples using a resistive magnet and superconducting magnet respectively. Traces in **a** and **b** are shifted vertically for clarity. **b** $H - P$ phase diagram showing all anomalies (red solid symbols). $H_1$ to $H_7$ indicate the sub-1/8 anomaly and the magnetization plateaus at ambient pressure; $H_0$ indicates the low-field anomaly. The blue and light blue symbols represent the 1/8, 1/4, 1/3 plateaus extracted from previous ambient-$P$ measurements in ref. [6] and ref. [12], respectively. The red open symbols indicate the splitting of the low-field anomaly at higher $P$.

phase, at 1.9 GPa and 2.3 GPa, we still can identify two anomalies in this field range, although they are now much weaker and shifted slightly to even lower fields. It is tempting to assign these two anomalies at these high pressures (1.1 GPa, 1.9 GPa, and 2.3 GPa) as extensions of the $H_6$ (1/4 plateau) and $H_7$ (1/3 plateau) seen at $P = 0$. However, we caution that the fate of the magnetization plateaus at higher pressure needs to be understood first. Indeed, as we show below, the real physical picture is much more complicated than a simple extension from the ambient pressure results. In fact, some of these anomalies actually signal previously unobserved phases that only appear at high pressure and high field, such as a 1/5 plateau phase and a $10 \times 2$ supersolid phase.

The interpretation of the data obtained at lower fields also requires some care. First, at $P = 0$, we identify three anomalies at $H_3$, $H_4$, and $H_5$, between the expected 1/4 and 1/8 plateaus (Fig. 1a and Supplementary Fig. 2a). Here, NMR measurements

have found evidence for 2/15 and 1/6 plateaus[11] for $H\|c$. After accounting for the g-factor difference between $H\|c$ and $H\|ab$, we find that two of the anomalies ($H_3$ and $H_5$) are located at fields consistent with the onsets of the 2/15 and 1/6 plateaus[11] (Fig. 1b). For $H_4$, it is likely associated with the 1/7 anomaly, the possible trace of an intermediate plateau[14] stabilized by factors such as inter-layer coupling.

At even lower fields, we identified $H_1$ and $H_2$ at $P = 0$ (Fig. 1a and Fig. 1b). Here, at ambient pressure and the field expected for the beginning of the 1/8 plateau[6], we consistently observe the weak feature marked as $H_2$ ($H\|ab$) (Fig. 1a and Supplementary Fig. 1). The identification of $H_2$ anomaly as the 1/8 plateau is also supported by comparison with the results for $H\|c$ after proper g-factor correction [refs. 6,9,10, also see Supplementary Fig. 1]. Below the onset of 1/8 plateau at $H_2$, we find a pronounced sub-1/8 anomaly at $H_1$, which seems to only appear for the $H\|ab$ orientation, and which corresponds to the large jump in magnetization that was reported in early studies[6] but not studied in detail. It has been suggested that any anomaly in this field range might be a hallmark of a higher order (e.g., 1/9 or 1/10) plateau[6,9,10]. The anisotropic behavior might suggest some role of the Dzyaloshinskii–Moriya (DM) interactions[14,42], which could stabilize or destabilize certain plateaus for different field orientations. However, as shown in ref. [14], the higher-order plateaus are much less stable compared to the 1/8 plateau, and the experimental results for higher-order plateaus so far are also not conclusive. It is thus highly unlikely that the rather weak DM interaction in $SrCu_2(BO_3)_2$ ($J_{DM}/J = 0.03 \sim 0.04$, refs. 14,42) could enhance the higher-order plateaus to such a degree. An alternative explanation is that the pronounced anomaly at $H_1$ corresponds to the transition between the single singlet condensation and the condensation of the bound states of triplets, which is theoretically expected to occur before the bound states crystallize at the 1/8 plateau. This behavior is more pronounced for $H\|ab$ than for $H\|c$, likely because the small separation between the two field scales is more apparent with the smaller g-factor along the $a$ and $b$ axes than along $c$.

At even lower fields, we observe the emergence of an anomaly near 7 T (Fig. 1a, right panel), which we refer to as $H_0$, only in the pressure range where plaquette state appears. At 2.2 GPa, this anomaly further splits. It is interesting to note that the magnetic energy scale of this anomaly is comparable to the low-energy excitation mode observed by inelastic neutron scattering in the plaquette state[25], albeit without observing the subsequent splitting at the higher pressure. Structure factor measurements of this low-energy mode suggested that the ground state is a full plaquette featuring diagonal bonds[25]. As shown below, our numerical results show that the splitting corresponds to a hidden AFM state, which is possibly connected adiabatically to the AFM ground state observed by heat capacity measurements above 2.5 GPa. This is consistent with the expectation that the AFM phase is favored at higher $T$, $H$, and $P$ where entropy is increased[31].

In Fig. 1b, we show the characteristic magnetic fields of all the anomalies as a function of pressure. Here, $H_2$, $H_6$, and $H_7$ correspond to the 1/8, 1/4, and 1/3 plateaus respectively at ambient pressure; $H_1$ is the sub-1/8 anomaly that signals the onset of the condensation of triplet-bound states; $H_3$, $H_4$, and $H_5$ are attributed to the intermediate magnetization plateaus as discussed above; $H_0$ represents the low-field anomalies that appear above 1.7 GPa. As we will show below, these characteristic fields constitute a rich phase diagram containing a variety of spin superstructures.

Finally, we have also investigated the effect of Mg dopants in the system (Supplementary Note 1 and Supplementary Fig. 10) and found that the results can be consistently explained by the impurity-induced spin structures that we established for ambient pressure[40].

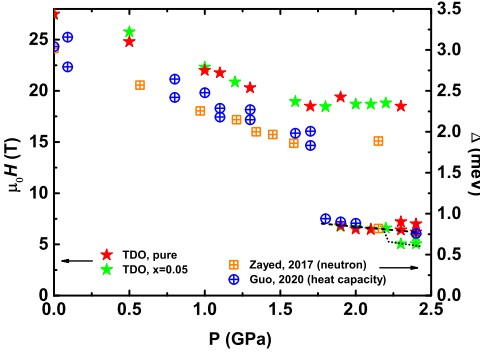

**Fig. 2 $H - P$ phase diagram of the sub-1/8 anomaly and the LE mode.** The $H - P$ phase diagram from our TDO measurements is compared with the $\Delta - P$ phase diagram established by neutron scattering[25] and heat capacity[27]. (Left axis) $\mu_0 H$ vs. $P$; Red and green stars are characteristic fields from the TDO results for the $x = 0$ and $x = 0.05$ samples. (Right axis) $\Delta$ vs. $P$; Orange squares and blue circles are the spin gap values reported by studies of neutron scattering[25] and heat capacity[27], respectively. Similar pressure dependence are observed for $\mu_0 H$ vs. $P$ and $\Delta$ vs. $P$.

We note that our low-field results are consistent with previous results in the entire pressure range. At ambient pressure, SrCu₂(BO₃)₂ has a 3 meV gap separating the spin singlet ground state and the triplet excited state[43]. Applying pressure within the dimer phase suppresses this gap[24], but it does not completely close before entering the plaquette state[25]. Inelastic neutron scattering measurements within the plaquette phase found the emergence of a low-energy mode along with a slight hardening of the triplon gap[25]. The pressure dependence of the former was tracked via heat capacity and was found to decrease with increasing pressure[27]. The spin gap can also be suppressed by the Zeeman mechanism, where the lowest excited state is brought down in energy by the application of the magnetic field. Interestingly, when plotted in the same figure, as shown in Fig. 2, the pressure dependence of some of the characteristic fields [$\mu_0 H_1(P)$ and $\mu_0 H_0(P)$] and that of the spin gap [$\Delta(P)$] measured by neutron scattering and heat capacity measurements show similar behaviors. On the other hand, some notable differences of the two types of pressure dependence are also observed at $P \gtrsim 2.3$ GPa. Here, $\mu_0 H_0$ splits, signaling the emergence of the AFM state. Our observations thus provide a broader perspective for the evolution of the spin gap with pressure in this material.

Finally, we note that while the introduction of Mg doping does not qualitatively change the behavior of $\mu_0 H_1(P)$ and $\mu_0 H_0(P)$, the anomalies presaging the AFM state in $\mu_0 H_0(P)$ are shifted to lower energy compared to that in pure SrCu₂(BO₃)₂, though the doping dependence of this softening remains to be explored (see Supplementary Fig. 5 for $x = 0, 0.05$ data collected at 2.4 GPa, Supplementary Fig. 6 for $x = 0.02, 0.03$ data collected at 2.1 GPa, and Supplementary Note 1). In the spin-dimer phase, adding impurities has been found not to move the onset fields of the 1/n plateaus, although increased impurity concentration does soften the spin superstructures and enhances the probability of forming impurity pairs and impurity-induced triplet states[40]. This suggests that the superstructures of the triplet-bound states have excitation energies independent of impurity doping. As shown in Fig. 2, this similarity between the pure and doped cases extends into the plaquette phase, indicating that the triplon excitation is likewise insensitive to impurity doping. However, the impurity-driven shift in the low-field mode $\mu_0 H_0(P)$ noted above suggests that the dopants act to destabilize the plaquette phase and instead favor the AFM phase, which is perhaps not surprising as we will show below that the AFM phase is also favored at higher $T$, $H$, and $P$.

**iPEPS calculation results.** We have performed iPEPS simulations (Methods) of the SS model in a magnetic field, given by the Hamiltonian:

$$H = J \sum_{\langle i,j \rangle} \boldsymbol{S}_{\boldsymbol{i}} \cdot \boldsymbol{S}_{\boldsymbol{j}} + J' \sum_{\langle\langle i,j \rangle\rangle} \boldsymbol{S}_{\boldsymbol{i}} \cdot \boldsymbol{S}_{\boldsymbol{j}} - h \sum_i S_i^z, \qquad (1)$$

where $J$ and $J'$ are the intra-dimer and inter-dimer couplings, respectively, and the strength of the external magnetic field is controlled by $h$. At ambient pressure a ratio $J'/J = 0.63$ was determined from a fit to high magnetic field data[12].

Applying pressure leads to an effective increase of the ratio $J'/J$; however, the precise pressure dependence of $J'$ and $J$ is not known. Here we model the pressure dependence by linear functions for $J(p)$ and $J'(p)$, with a change of 5% in $J'$ between its value at ambient pressure and its value at the critical pressure $p_c = 1.8$ GPa [corresponding to $J'/J = 0.675$ (ref. [36])]. A change of 5% is also predicted from ESR data[26], and is close to the estimate (4%) obtained in a recent ab-initio study[29] (in contrast, a substantially larger value (17%) was found from fits to magnetic susceptibility data[25]). At ambient pressure, we use $J = 81.5$ K which lies in between previously predicted values[12,44] and yields good agreement with the onset of the 1/4 and 1/3 plateaus observed in experiments. The resulting slopes of the linear functions $J'(p)$ and $J(p)$ are $-1.43$ K/GPa and $-5.13$ K/GPa, respectively. In Supplementary Figs. 11 and 12 and Supplementary Table 1, we present alternative phase diagrams using different parameter sets, in order to illustrate the dependence of the phase boundaries on the various parameters. The boundaries change very little as long as the variation of $J'$ is only a few percent. The change by 17% deduced from fitting the susceptibility[25] would by contrast lead to a phase diagram whose boundaries depart significantly from our experimental data and thus can be discarded.

We focus in the following on some of the most prominent features in the phase diagram as a function of $J'/J$ (or pressure) and magnetic field. In particular, we concentrate on the phase boundaries of the magnetization plateaus and the supersolid phases at high magnetic fields, and the competing low-energy states at high pressure. The results, summarized in Fig. 3, are obtained for $D = 8$ which already provides an accurate estimate of the phase boundaries (e.g., the relative error on the phase boundaries of the 1/4 and 1/3 plateaus is <2% compared to the results extrapolated to the infinite $D$ limit[12]).

At high fields (up to 45 T) the dominant phases are the 1/4 plateau, the 1/3 plateau, and a 1/3 supersolid phase[12]. A supersolid phase simultaneously breaks the translational symmetry and the U(1) symmetry associated with the total $S^z$ conservation. The 1/3 supersolid exhibits the same translational symmetry breaking pattern as the 1/3 plateau state, but with additional spin components in the transverse direction, reflecting the broken U(1) symmetry. The 1/4 plateau has a finite extent up to $J'/J = 0.675(5)$ after which the intermediate field region is dominated by the 1/3 supersolid phase. The 1/3 plateau remains stable over the entire range of $J'/J$ considered here. Below the 1/4 plateau at ambient pressure there is a sequence of small magnetization plateaus (crystals of triplet-bound states)[14], denoted as "intermediate plateaus" in Fig. 3, which are stable up to $J'/J = 0.675(5)$. We also add a characteristic line indicating a lower bound for the onset of the 1/8 plateau. This line is obtained by intersecting the energy of the 1/8 plateau with that of the 1/9 plateau, a plateau which is however probably unstable towards a condensate of spin-2 bound states (see above).

At intermediate fields and ~$J'/J = 0.68$, we find a 1/5 plateau that has not been observed previously (we call it high-$P$ 1/5 plateau); and it is different from the 1/5 plateau made of localized triplet-bound states appearing at smaller $J'/J$ (or lower pressure,

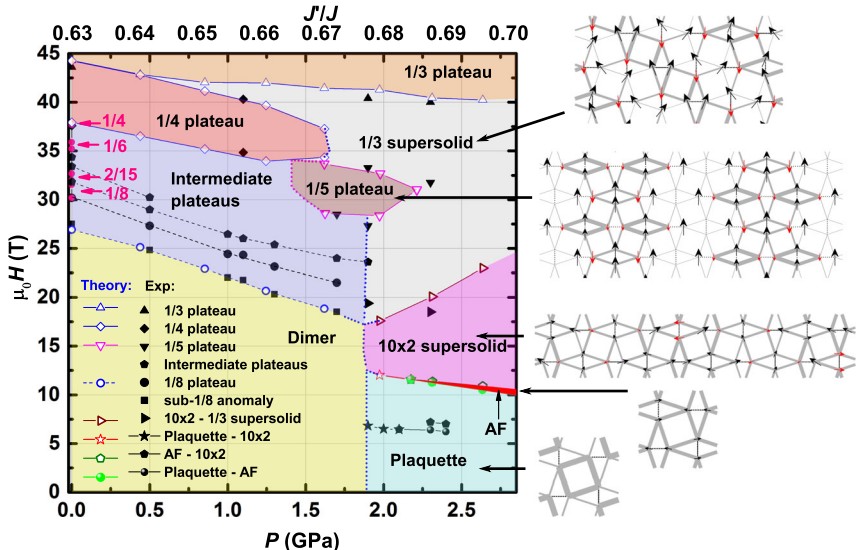

**Fig. 3 $H - P$ phase diagram (theory vs. experiment).** Black symbols and lines correspond to anomalies found in experiments, the colored symbols and lines are based on iPEPS results ($D = 8$). The corresponding coupling ratios $J'/J$ are shown on the top axis. The colored phase regions are determined by the iPEPS data points. Dotted lines are a guide to the eye. The experimental data and the iPEPS data agree well, except for two places: near the 1/8 plateau and the Plaquette-AFM-10 × 2-supersolid transitions. The iPEPS calculation does not capture the sub-1/8 anomaly which is very pronounced for $H\|ab$ but invisible for $H\|c$ (ref. [6]) because of the isotropic nature of the standard SS model. On the right-hand side, typical spin patterns of the phases at high pressure are drawn. The size of the spins scale with the magnitude of the local magnetic moment, where black (red) arrows point along (opposite to) the external magnetic field. The thickness of the gray bonds scales with the local bond energy (the thicker the lower the energy).

ref. [14]). The spin structure of this plateau exhibits a stripe pattern parallel to one set of dimers, as shown in Fig. 3, where along each stripe a strong dimer triplet alternates with a pair of weaker dimer triplets. We will discuss the physical origin of this rather unusual structure in the next section.

At ambient pressure as well as at 1.1 GPa, we find a very good agreement between the phase boundaries of the 1/4 plateau and the critical fields of the anomalies found in experiments. At 1.9 GPa and 2.3 GPa, the anomalies at 40 T are close to the iPEPS phase boundary of the 1/3 plateau, and the anomalies at 34 T are in good agreement with the upper edge of the high-$P$ 1/5 plateau. At 1.7 GPa and 1.9 GPa, we also identify two anomalies consistent with the lower edge of the high-$P$ 1/5 plateau. All these features are thus well captured by the standard SS model and by the simple model for the pressure dependence of $J$ and $J'$ used here.

Finally, we turn our focus to the low-field region at high pressure. Above the empty plaquette (P) phase in zero and small fields, we find a narrow partially polarized AFM phase, and a $10 \times 2$ supersolid state (obtained in a $10 \times 2$ unit cell; hence the name), followed by the 1/3 supersolid and 1/3 plateau phases. The corresponding spin patterns are displayed in Fig. 3, and examples of magnetization curves are shown in the Supplementary Figs. 7 and 8. Note that the pattern of the $10 \times 2$ supersolid phase is different from that of the stripe phase reported in ref. [45]. Interestingly, the anomalies at 1.9 GPa and 2.3 GPa just under 20 T in experiments lie close to the phase boundary between the $10 \times 2$ and 1/3 supersolid phase. As we shall see below, the $10 \times 2$ supersolid phase can be seen as a descendant of the unusual 1/5 plateau.

The narrow AFM region, which vanishes around $J'/J \approx 0.686$ and which becomes broader with increasing $J'/J$, is qualitatively compatible with the splitting of the two anomalies observed at low fields in experiments. However, quantitatively we find that these phase boundaries occur at higher fields than in experiments. We believe the main reason for this discrepancy is the lack of inter-layer coupling in our model, which is of order of 0.09$J$ (ref. [18]) and which is expected to enhance the stability of the AFM

phase compared to the plaquette state[18] leading to a shift of the phase boundary to smaller critical fields. Additionally, Dzyaloshinskii–Moriya (DM) interactions[17–21] of order of a few percent of $J$ may affect the location of the phase boundaries. We note finally that the empty plaquette state is different from the full-plaquette state implied by experiments, which can be obtained using a deformed Shastry–Sutherland model[28,46] that includes two types of intra- and inter-dimer interactions. These modifications of the model may also affect the magnetization process, particularly at low fields, where the narrow AF phase is energetically closely competing with the plaquette and the $10 \times 2$ phase. We stress, however, that the high-$P$ 1/5 plateau and the $10 \times 2$ supersolid phase remain relevant ground states also in the deformed model (Supplementary Fig. 13). In fact, they tend to be further stabilized by the deformation, a logical tendency since they correspond to descendants of the full-plaquette state, as we discuss in the following sections.

**Nature of the high-$P$ 1/5 plateau and $10 \times 2$ supersolid.** As stated in the Introduction, the high-field plateaus can be thought of as Wigner crystals of $T_1$ magnetic particles, while the lower plateaus are better interpreted as Wigner crystals of spin-2 bound states of such $T_1$ particles[14]. The resulting structures are very simple to visualize. The high-field plateaus build diagonal stripes (in a geometry where dimers are horizontal or vertical), while the Wigner crystals of spin-2 bound states consists in putting the bound states as far as possible from each other.

The two phases discovered in the present paper, the high-$P$ 1/5 plateau and the $10 \times 2$ supersolid, are completely different and cannot be understood in these terms. Since the $10 \times 2$ supersolid can be understood as its descendant, let us first concentrate on the high-$P$ 1/5 plateau. Its main properties are, (i) The stripes are vertical, and not diagonal; (ii) The state is not a simple Wigner crystal of $T_1$ particles, but half the magnetic particles are delocalized over two dimers; (iii) This gain of kinetic energy cannot be achieved with the mechanism that explains the other

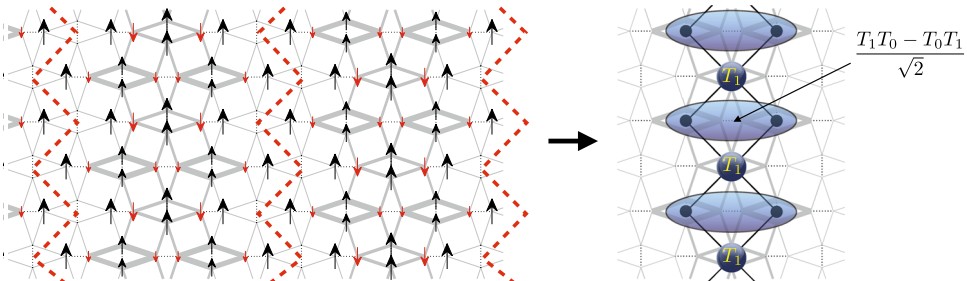

**Fig. 4 Nature of the high-$P$ 1/5 plateau.** The spin structure consists of vertical stripes of partially polarized full plaquettes which are separated by vertical singlets along the red dashed line. Each stripe can effectively be described by a spin-1 diamond chain in a magnetic field at filling 2/3, in which polarized $S = 1$ spins ($T_1$) and dimer triplets, made of an antisymmetric combination of $T_1$ and $T_0$, are alternating.

plateaus. Indeed, in order for a triplet to jump on a horizontal next-nearest neighboring dimer, this neighbor must be occupied by a non-magnetic particle called $T_0$, a triplet with zero magnetization along the field. In fact, the necessity to include $T_0$ particles in the description of this plateau has been proven in the context of a similar plateau found in a thin-tube version of the SS model made of two orthogonal dimer chains[47].

Interestingly, this plateau can be easily understood as a descendant of the full-plaquette phase. In the full-plaquette phase, the lattice symmetry is broken, and half the dimers, say the horizontal ones, are singlets, while the other ones are triplets. These triplets build effective Haldane spin-1 chains[46], leading to the full-plaquette phase (called for that reason the Haldane phase in earlier papers). Similarly, in the high-$P$ 1/5 plateau, all horizontal dimers are singlets, but a subset of vertical dimers along vertical paths also form dimer singlets, effectively cutting the system into vertical stripes, Fig. 4. Quite remarkably, each vertical stripe is equivalent to a well known model of 1D quantum magnetism, the spin-1 diamond chain, each spin-1 corresponding to a triplet on a vertical dimer. The physics of this model is very simple to understand. It can be seen as an alternating set of single spins and dimers and because of the symmetric position of the dimers with respect to its neighboring single spins, the total spin of each dimer is a good quantum number. As a consequence, the magnetization process can be shown to lead to three plateaus at 1/3, 2/3, and 1 depending on whether the total spin of each dimer is a singlet (1/3), a triplet (2/3), or a quintuplet (1). The high-$P$ 1/5 plateau corresponds to the intermediate plateau of this diamond chain, and the wave-function of the "delocalized" particle is an antisymmetric combination of a $T_1$ and a $T_0$ on the vertical dimers over which it is delocalized, as shown on the right in Fig. 4. Note that the identification of the full-plaquette phase as the parent state of this plateau is further supported by the strong $J'$ bonds within the stripes, which are clearly those of full plaquettes.

Finally, the $10 \times 2$ supersolid phase can be obtained from the high-$P$ 1/5 plateau by an alternating rotation of the magnetization of successive stripes clockwise or counterclockwise by 90°, and by adding some magnetic particles between the stripes, see Supplementary Fig. 9. We note that, because the $10 \times 2$ supersolid is not obtained by just a small rotation of the spins of the high-$P$ 1/5 plateau, we a priori do not expect it to be adjacent to the high-$P$ 1/5 plateau phase.

## Discussion

In summary, our results at high magnetic field and high pressure using TDO magnetization measurements and iPEPS calculations elucidate the very rich multi-dimensional ($H - T - P$) phase diagram of SrCu$_2$(BO$_3$)$_2$. One notable feature of our phase diagram is the vast region occupied by multiple supersolid phases in the high-$H$ and high-$P$ regime (Fig. 3). These phases are the

magnetic analogs of the supersolid which were originally predicted to appear in $^4$He (refs. [48–51]) and realized only in optical lattices[52,53]. While the spin analogs of supersolids have been confirmed and discussed extensively in 1D and 2D frustrated magnetic models[54–56], and the possibility of realizing such phases in SrCu$_2$(BO$_3$)$_2$ near its magnetization plateaus was suggested initially two decades ago[15] and was subsequently bolstered by numerical studies[12,16,57], direct experimental identification of the supersolid phase space has been less clear and the evidence has been confined only to the ambient pressure[12]. This is in part because the volume of parameter space over which supersolids might be observed is expected to be very small at ambient pressure[12,14] due to the competition between the correlated hopping and the repulsive interaction of the triplets. The correlated hopping term is strongly suppressed at low $J'/J$ where the coupling is small. With increasing pressure, the correlated hopping is enhanced, which increasingly favors the supersolid as the ground state as opposed to the crystal of (bound states of) triplets. While the numerical studies agree on these broad contours, there has not been consensus on the exact phase diagram[12,15,57] because of the lack of experimental evidence. Moreover, the magnetization anomalies associated with the supersolids are close in energy with those at magnetization plateaus[12,14,28], and it is difficult to clearly differentiate the two experimentally without studies of their pressure dependence. In this regard, our results establish the extent and phase boundaries of the supersolid phases in SrCu$_2$(BO$_3$)$_2$. The large region of stable supersolid phases provides valuable opportunities for further studies of the properties of supersolids. For example, it would be interesting to understand the phase transitions between the supersolid phases and supersolid to solid.

Meanwhile, several phases are reported, including an intermediate-field AFM state, and two highly exotic structures, a high-$P$ 1/5 plateau and a related $10 \times 2$ supersolid. We have argued that the latter two cannot be understood in terms of Wigner crystals of $T_1$ magnetic particles (or bound states thereof), but that they instead correspond to descendants of the full-plaquette state, which at zero field is energetically very close to the empty plaquette state in the Shastry–Sutherland model[28], and which is believed to be realized in SrCu$_2$(BO$_3$)$_2$.

The physics of SrCu$_2$(BO$_3$)$_2$ under pressure seems to be dominated by a tendency towards an orthorhombic distortion that stabilizes the full-plaquette phase and its descendants. These findings make the investigation of the symmetry of the intermediate plaquette phase more relevant than ever, and suggest that the transition between this phase and the AFM phase should be revisited in the context of a model that includes this tendency. Ideally this should be done in the context of a model where the orthorhombic distortion is spontaneously broken, but this might require including the coupling to the lattice, a formidable challenge for numerical methods. This is left for further investigation.

Beyond $SrCu_2(BO_3)_2$, we note that the freedom to tune across energy scales that are relevant to the underlying interactions is crucial in research of strongly correlated systems, and often requires the use of extreme experimental conditions. In that respect, our experimental results provide a road-map for exploring correlated matter in extreme environments of low temperature, high magnetic field, and high pressure. Indeed, supplemented by the powerful iPEPS simulations, our results reveal the extent of the phase diagram of a prototypical correlated system that would otherwise be impossible to access. Although the TDO technique itself has been well developed, its high adaptability in different sample environments and high sensitivity to detect magnetization changes have only been exploited recently, and the present work goes one step forward by combining high field and high pressure. Similar efforts in redefining the capability of other existing experimental techniques could be a rewarding direction in future exploration of strongly correlated matter.

Moreover, our results not only demonstrate the unique power of TDO in probing materials in combined extreme conditions of field and pressure, but also predict the stabilization in a range of different types of plateaus beyond those already identified (Wigner crystals of triplets and Wigner crystals of bound states of triplets). These phases can be interpreted as Mott insulating phases of hard-core bosons, the role of the magnetic field being played by the chemical potential. This unexpected finding might thus have implications in other contexts where interacting bosonic models are relevant.

## Methods

**Sample synthesis and characterization**. High quality single crystal samples of both $SrCu_2(BO_3)_2$ and $SrCu_{2-x}Mg_x(BO_3)_2$ ($x = 0.02, 0.03$, and $0.05$) were grown by the optical floating zone technique using self-flux, at a growth rate of 0.2 mmh$^{-1}$ in an $O_2$ atmosphere[40,58]. The quality of the samples and the doping levels of the Mg-doped samples are confirmed in our previous study using the same batch of samples[40]. In ref. [40], the experimentally extracted doping concentrations using Curie–Weiss fit were found to match well with the nominal doping levels for $x = 0.02, 0.03$, and a gradual evolution of the magnetization behavior with increasing doping up to $x = 0.05$ was observed.

**Tunnel diode oscillator (TDO)**. Cylinder-shaped crystals of ~ 2 mm in length and ~ 1 mm in diameter are used for the TDO measurements. The samples are oriented using Laue diffraction so that the long edge of the sample is along the a-axis (or equivalently, b-axis). The sample was then placed inside a detection coil with inductance $L$, which is small enough to be inserted into a Copper-Beryllium piston pressure cell with a maximum pressure rating of 2.4 GPa. When mounting on the sample stage, the coil is oriented by hand such that the field is applied parallel to the a-axis (or b-axis) within ~ 5°. The coil with the sample inside and a capacitor is used to form an LC circuit. The changes in sample magnetization is reflected as the change in resonance frequency, which can be measured to high precision. The experiments were conducted at the dc field facility of the National High Magnetic Field Laboratory in Tallahassee, FL.

**Identification of the magnetization anomalies in the TDO data**. The extreme sensitivity of the TDO frequency to the magnetization change allows an accurate determination of the rich magnetic phase diagram of $SrCu_2(BO_3)_2$. At ambient pressure, the TDO anomalies have been identified as magnetization changes associated with the magnetization plateaus and other magnetic phases[13,40]. These anomalies also evolve under pressure, and we are able to track them up to 2.4 GPa.

Because the TDO frequency is related to the magnetization by $df/dH \propto -dM^2/d^2H$ (ref. [40]), a local minimum ("dip") and maximum ("bump") in $df/dH$ correspond to where $dM/dH$ changes the fastest. In principle, two types of behaviors are usually observed at a magnetization anomaly in experiments. First, $df/dH$ could appear with only a "dip", which corresponds to a slope change in $M(H)$. A typical example is shown in Fig. 1a for the 1.1 GPa trace near 40 T, which we identify as the exit of a 1/4 plateau. Except for this one example, $df/dH$ appear with a "dip" followed by a "bump". The midpoint between the "dip" and the "bump" corresponds to where the rate of change in $M$ is the highest, i.e., jump in magnetization. Therefore, for all the TDO anomalies, we identify the "dips", then the "bumps" when possible, and find their midpoint, which are used as the data points plotted in the phase diagrams. Typical examples are shown in Supplementary Figs. 2 and 3. For the 1/3 plateau, however, no "bumps" can be identified except for those at high pressures ($P > \sim 1.8$ GPa), due to the limitation in

available magnetic field. Therefore, we identify the exit of the 1/4 plateau and the onset of the 1/3 plateau with the field values of the "dips".

Moreover, the TDO signal is susceptible to the background interference because of its ultra high sensitivity. Therefore, we have only identified the TDO features that are repeatable for different samples in different magnet runs. As shown in Supplementary Figs. 2 and 3, the high-$P$ 1/5 plateau is identified for (1) two different $x = 0$ samples that are measured in two different magnet runs using a resistive magnet and a hybrid magnet respectively; (2) both the $x = 0$ and $x = 0.05$ samples. Meanwhile, the features should also have a clear trend with the pressure.

For $P = 0$, we also make comparison between our TDO data and the reported magnetization data[6] and NMR data[11]. The results agree well (Fig. 3 and Supplementary Fig. 1). Note that the NMR data in ref. [11] was conducted with $H\|c$. Therefore, we have converted the field values of the magnetization plateaus to those of $H\|ab$ by multiplying them with the g-factor ratio $g_{\|c}/g_{\|ab} = 2.28/2.04 = 1.12$ (ref. [40]). The obtained results agree well with our iPEPS calculation, as shown in Fig. 3. Examples of our detailed analyses are given in Supplementary Figs. 1, 2, and 3.

**Infinite projected entangled pair states (iPEPS)**. An iPEPS is a variational tensor network ansatz to represent 2D ground states in the thermodynamic limit[59–61], where the accuracy is systematically controlled by the bond dimension $D$ of the tensors. The ansatz consists of a unit cell of tensors, here with one tensor per dimer[36]. The optimization of the variational parameters is done based on an imaginary time evolution using the simple update approach[62], which provides good estimates of ground-state energies while being computationally affordable even for large unit cell sizes. For small unit cells with two tensors we further improved the results using the fast full update optimization[60,63,64] and we made use of the variational optimization[65] to create initial states at low $D$. The contraction of the infinite 2D tensor network is done by a variant[66,67] of the corner-transfer matrix method[68,69]. Further details on the iPEPS approach can be found in refs. [36,63,64].

## Data availability

The data that support the findings of this study are available within the paper and the Supplementary Information. Additional data related to this paper may be requested from the authors.

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

## Acknowledgements

We are grateful to Casey Marjerrison for her assistance with crystal growth activities at the early stages of this project. A portion of this work was performed at the National High Magnetic Field Laboratory, which is supported by the National Science Foundation Cooperative Agreement No. DMR-1157490 and DMR-1644779, the State of Florida and the U.S. Department of Energy. Z.S., S.D., W.S., and S.H. acknowledge support provided by funding from the Powe Junior Faculty Enhancement Award, and William M. Fairbank Chair in Physics at Duke University. D.M.S. and T.F.R. acknowledge support from US Department of Energy Basic Energy Sciences Award DE-SC0014866. P.C. and F.M. acknowledge the support provided by Swiss National Science Foundation and the European Research Council (ERC) under the European Union's Horizon 2020 research and innovation programme (grant agreements No 677061 and No 101001604).

## Author contributions

Research conceived by S.H.; Single-crystal SCBO samples grown by H.A.D. and S.H.; High-field measurements performed by Z.S., S.D., W.S., D.G., and S.H., and analyzed by Z.S., S.D., D.M.S., T.F.R., and S.H.; iPEPS calculations performed by P.C. and F.M.; manuscript written by Z.S., P.C., F.M., D.M.S., T.F.R., and S.H.; all authors commented on the manuscript.

## Competing interests

The authors declare no competing interests.
