## [Peer Review File · Nature Communications]

REVIEWER COMMENTS

Reviewer #1 (Remarks to the Author):

SrCu₂(BO₃)₂ has been known as a realization of the frustrated 2D Shastry-Sutherland spin model and extensively studied over past twenty years mainly along two different directions. One is the remarkable sequence of multiple magnetization plateaus, which are induced by applying magnetic fields to the dimer singlet ground state and understood as crystallization of triplets or bound triplet pairs. The other direction is the quantum phase transitions induced by applying high pressure, that changes the ratio of intra- to interdimer exchange couplings. In this paper, the authors report surprisingly rich phases of SrCu₂(BO₃)₂ including a new plateau and supersolid phases obtained in the multiple extreme conditions combining both high magnetic fields and high pressures. For this purpose, they used sophisticated experimental and theoretical techniques, namely the tunnel diode oscillator for precise measurements of magnetization anomalies and iPEPS calculations for identification of the phases. In my opinion, this work has opened a new route to explore even more exotic quantum states in this remarkable material than known to date. Therefore, I strongly recommend publication of this work in Nature Communications once the authors have considered the following questions and comments.

1. The authors assert that the sub-1/8 anomaly at $H \sim 27.5$ T is caused by condensation of $S=2$ bound triplet pairs (pages 7 and 8). Are there any experimental evidence for this interpretation? Since the sub-1/8 anomaly has been seen only for the field in the ab plane and absent for $H // c$, it seems to me more reasonable to ascribe its origin to anisotropic interactions such as DM interaction. The condensation of bound triplets in the SS model should be an isotropic phenomenon when the fields are multiplied by the g -values.

2. I cannot find any “anomalies at 1.9 GPa and 2.3 GPa around 21.5 T in experiments” (second paragraph of page 11) either in Fig 1 or Fig. 3.

3. The observation of descendant relation among the full praquette, 10×2 supersolid, and $1/5$ plateau phases is very interesting. Then I would expect the transition directly from 10×2 supersolid to $1/5$ plateau. However, the iPEPS result shows a wide intervening region of $1/3$ supersolid phase in between. Is there any simple explanation for this ?

4. Related to the previous question, since the full praquette phase does not appear in the SS model, the lowest field phase obtained by iPEPS above 1.9 GPa in Fig. 3 must be the empty praquette phase.

Then It seems to me logically inappropriate to apply the above argument of descendant relation to interpret the results in Fig. 3.

Reviewer #2 (Remarks to the Author):

The authors present a study of the material $\text{SrCu}_2(\text{BO}_3)_2$ using TDO (tunnel Diode Oscillator) technique at high pressures, high magnetic fields and low temperatures. The experimental study is accompanied by a numerical analysis based on iPEPS calculations.

By combining their experimental and numerical results the authors propose a field- pressure phase diagram for $\text{SrCu}_2(\text{BO}_3)_2$ exhibiting many different magnetic phases, some of which were previously observed in that compound while others are novel.

Experiments at high pressure and furthermore, experiments combining high pressures, high magnetic fields and low temperatures are notoriously difficult. With TDO measurements, the authors present a phase diagram that covers a large range, with pressures from 0 to about 2.5 GPa and magnetic fields from 0 to 45 T. It also contains relatively precise boundaries between the different proposed phases.

As such, the study is of particular interest for quantum magnetism as it provides novel information about an important compound and its theoretical counterpart, the Shastry-Sutherland model.

In the TDO measurements some of the features are easy to spot while some others require more analysis and are harder to see. The later depend on consistency checks that the authors have made and that cannot be easily checked by peer review. However, in my opinion, the authors have provided reasonably sufficient justification about the location of the different features mentioned in the manuscript.

The details of iPEPS calculation are not part of my expertise and I cannot comment on those.

The references given are appropriate.

I would therefore in principle recommend it for publication in Nature communications provided the issues raised below are answered.

Main concerns and questions

1. The numerical calculations make use of the pressure dependence of J and J' . However as mentioned in the text these are not well known. The ab-initio paper in Ref 26, itself contains several values and references. The $J(p)$ and $J'(p)$ chosen by the authors seem not to be any of the previously published ones. $J(p)$ is not explicitly given and $J'(p)$ is taken to vary by 3% from 0 to 1.8 GPa. First, can the slope of $J(p)$ be given. Second, this raises a question about how robust the numerical analysis is with respect to the “choice” of the pressure dependence of the exchange parameters.

The authors should address this issue and explain or show how changes in the initial values of J and J' or in their slope would affect their results. In particular does the “match” between calculations and experiment in figure 3 strongly depend on those choices?

2. Comparison neutron vs TDO: In Figure 2, the TDO measurements are systematically higher (by 0.25 to 0.5 meV) than the spin gaps reported by neutrons and specific heat measurements in the dimer phase. In the plaquette phase however there is no shift for the low energy modes while the shift remains for the higher energy mode. Could the authors address and discuss this issue?

Is that a due to a difference in dispersion? Is that a systematic of the TDO measurement compared to other techniques. Why does the systematically higher TDO values suddenly match the neutron and specific heat values for the lower modes of the plaquette?

Minor concern

3. a) In page 7: “H1 is the sub-1/8 anomaly that signals the onset of the condensation of triplet bound states”. Is that claim based on the current analysis and can some justification be given, or is it based on a previous work in which case a reference should be given? There is not much further discussion in the main manuscript, but in the supplementary material, it appears that it is a new interpretation. Can some more quantitative justification be provided?

b) Would other such “sub-plateau” anomalies be expected for the other plateaus?

4. Doped material. a) Do the indicated percentage correspond to the input composition during crystal growth or to the final result obtained. b) Were X-rays performed to verify the doping levels and the fact that the Mg impurities did actually go into the Cu positions? Nothing is said about sample characterization in the methods section.

5. In Supplementary figures 5 and 6. There is no big shift in the dip (and bump) position from $x=0$ to $x=0.03$ doping. At $x=0.05$ there is a sudden jump. The first dip, for instance, stays close to 6.0 GPa all the way to $x=0.03$ and then jumps to ~ 4.8 GPa for $x=0.05$. The authors should discuss this peculiar behavior.

6. a) Two orientations were measured: with H parallel to c and H perpendicular to c (i.e in the a-b plane). In the second case was the orientation systematically the same (for example H // to a) or did it change from one run to the next?

b) Should one expect a difference in magnetization or in df/dH when, for instance, H//a compared to when H is in the a-b plane at 45 deg between a and b?

Clarification needed

7. Not clear in text in page 6:

“similar anomalies”. Does this refer to H6 and H7 only or all of H1 to H7?

“this field range”, does it refer to the field for H6 and H7 only?

8. Page 8 “ We first focus on H1, ...” the structure of the presentation is not very clear. One would expect then a focus on the other features, but that is not how the text continues.

9. In page 11-12, when discussing the discrepancies between calculation and experiment, the authors invoke the 3D and DM couplings and then possible structural differences related to open vs full plaquettes. All of these are acceptable as possible sources of discrepancies. However, it is not clear at the end of the paragraph why those should be important “particularly at low fields”.

10. It would be good for the reader, in my opinion, to 1) explain in the text why some of the phases correspond to a “supersolid” state. For instance, what characteristics of a supersolid do these particular spin arrangements display? And 2) justify or explain the naming used such as: “10x2” and “1/3” supersolids.

11. Supplementary Fig S4. The quality of the data at 0.3K is much better than the data at 0.44K. Is that uniquely due to the physics or was the 0.44K data (and higher T) taken with lower statistics? It would be good to give this information in the supplementary text.

12. Supplementary Fig 5-6. Is the vertical scale the same for all panels?

Reviewer #3 (Remarks to the Author):

The manuscript by Shi et al combines experimental and theoretical efforts to explore the magnetic phase diagram of $\text{SrCu}_2(\text{BO}_3)_2$ under extreme conditions: sub-Kelvin temperatures, high pressures and high field. The anomalies observed in the tunnel diode oscillator (TDO) experiment are then used to establish the phase boundaries. By comparing to the iPEPS simulations, the authors claim the discoveries of a few new phases, in particular the 10x2 supersolid and a 1/5 plateau.

While it is always interesting to find novel magnetic states, for example the 10x2 supersolid and the 1/5 plateau, I am not fully convinced that this work meets the criteria of validity and novelty to publish under Nature Communications. My major concerns are list below:

1. As the authors admitted themselves (last paragraph in “iPEPS calculation results”), the model they are using does not even correctly reproduce the “full”-plaquette ground state at high pressure and zero field. Consequently, the theoretical findings at high pressure are sitting on a wrong base. This naturally explains their discrepancy in matching the theoretical and experimental data at pressures larger than 1.9GPa. Consequently, I am not convinced that the 10x2 supersolid phase and the narrow AF phase are relevant to the real material.

2. In Fig. 3, There is only one experimental data point near the upper phase boundary of the 10x2 supersolid phase, and there is no matching experimental/numerical data point at the lower phase boundary. This again questions the validity of the 10x2 phase.

3. For the 1/5 plateau and 1/3 supersolid, there are only limited experimental data points for the phase boundaries (three~four for 1/5 plateau and two~three for 1/3 supersolid). The interpretation of these phase are mostly relying on the model which could not even reproduce the zero phase full plaquette phase, and there is no direct experimental evidence.

4. Even if the numerically discovered new phases are established, I am not convinced yet that these results are of high significance to the field that qualifies publication in a high profile journal like the Nature Communication. Besides showing novel magnetic structures, the authors should provide evidence that such findings have significance either on the conceptual or at the application level.

Reply to the Reviewers

We are grateful to the Reviewers for their insights and for appreciation of the importance of our work. The constructive suggestions by the Reviewers on content and clarity have prompted us to significantly improve our manuscript. We have now revised our manuscript to address the Reviewers' comments. The changes are summarized below (comments in blue, replies to comments in black):

Reviewer #1

SrCu₂(BO₃)₂ has been known as a realization of the frustrated 2D Shastry-Sutherland spin model and extensively studied over past twenty years mainly along two different directions. One is the remarkable sequence of multiple magnetization plateaus, which are induced by applying magnetic fields to the dimer singlet ground state and understood as crystallization of triplets or bound triplet pairs. The other direction is the quantum phase transitions induced by applying high pressure, that changes the ratio of intra- to interdimer exchange couplings. In this paper, the authors report surprisingly rich phases of SrCu₂(BO₃)₂ including a new plateau and supersolid phases obtained in the multiple extreme conditions combining both high magnetic fields and high pressures. For this purpose, they used sophisticated experimental and theoretical techniques, namely the tunnel diode oscillator for precise measurements of magnetization anomalies and iPEPS calculations for identification of the phases. In my opinion, this work has opened a new route to explore even more exotic quantum states in this remarkable material than known to date. Therefore, I strongly recommend publication of this work in *Nature Communications* once the authors have considered the following questions and comments.

We thank the Reviewer for the positive assessment and the appreciation of our results, and for the strong recommendation of publication of our paper in *Nature Communications*. We also appreciate the insightful questions and comments by the Reviewer. As described below, we have carefully considered them and revised our manuscripts in response to the Reviewer's comments. We hope the Reviewer will find our response satisfactory and our revised manuscript suitable for publication in *Nature Communications*.

1. The authors assert that the sub-1/8 anomaly at H₁~27.5 T is caused by condensation of S=2 bound triplet pairs (pages 7 and 8). Are there any experimental evidence for this interpretation? Since the sub-1/8 anomaly has been seen only for the field in the ab plane and absent for H // c, it seems to me more reasonable to ascribe its origin to anisotropic interactions such as DM interaction. The condensation of bound triplets in the SS model should be an isotropic phenomenon when the fields are multiplied by the g-values.

We thank the Reviewer for the comment. The sub-1/8 anomaly is indeed an interesting observation that we believe has been mostly omitted in the literature. In fact, this anomaly was first noted in Ref. 5 [Onizuka, K., *et al.* J. Phys. Soc. Jpn. 69, 1016 (2000)], where it was found that the g-factor normalized perpendicular magnetization and parallel magnetization overlap in the entire field range except for the region near 30 T. Ref. 5 discussed the possibility of an anisotropic 1/10 plateau but did not reach a definite conclusion. To the best of our knowledge, this question has not been addressed in any subsequently published studies.

Therefore, we have carefully considered possible scenarios responsible for the sub-1/8 anomaly (in Supplementary Note 1 of our previous version of the manuscript). First, we agree with the Reviewer that anisotropic interactions such as DM interaction have to be important because of the clear anisotropic behavior of the perpendicular magnetization and parallel magnetization. However, the DM interaction J_{DM} , being much smaller than the exchange interaction J ($J_{DM}/J = 0.03 \sim 0.04$, see Ref. 13 and references therein), could not directly account for such a large sub-1/8 anomaly by itself. It is more likely that the DM interaction alters the stability of some steps in the magnetization process.

In this regard, we think plateaus such as the 1/10 plateau proposed by Ref. 5 are highly unlikely. Some of us (P.C. and F.M.) have shown in Ref. 13 that a high density of plateaus lies energetically very close below the 1/8 plateau and are much less stable compared to the 1/8 plateau. Experimental evidence so far for the higher order plateaus are also inconclusive. Although anisotropic interactions such as intra- and inter-dimer DM interactions could stabilize and destabilize certain plateaus, the sub-1/8 anomaly is too strong compared to the 1/8 plateau to be induced by any realistic values of DM interaction strength.

On the other hand, experimental signatures of condensation of $S=2$ bound states have been elusive and have not been intentionally searched for, even though it has to occur before the $1/8$ plateau where the $S=2$ bound states crystallize (see Ref. 13). It is, therefore, likely that the condensation of $S=2$ bound states occurs at a field very near the $1/8$ plateau and the two cannot be distinguished in the perpendicular magnetization. In the parallel magnetization, however, the separation between the condensation of $S=2$ bound states and the $1/8$ plateau could be enhanced by the larger g -factor and by the anisotropic DM interaction.

In response to the Reviewer's question, and also a similar question by Reviewer 2, we have made the following changes:

(1) We have added a new reference (Ref. 42), where the strength of the DM interaction was measured. The new reference is cited in the revised text mentioned in (2).

(2) We have moved the discussion in Supplementary Note 1 to the main text, and expanded it. The revised text in pages 7-8 now reads (due to the length of the text, we refer the Reviewer to the revised manuscript for the full change):

"At even lower fields, we identified $H1$ and $H2$ at $P=0$ (see Fig. 1a and Fig. 1c)...

...This behavior is more pronounced for $H||ab$ than for $H||c$, likely because the small separation between the two field scales is more apparent with the smaller g -factor along the a and b axes than along c ."

(3) We reworded the first two sentences of the 2nd paragraph in page 8 for a more smooth transition:

"At even lower fields, we observe the emergence of a new feature near 7 T ... At 2.2 GPa, this new feature further splits."

2. I cannot find any "anomalies at 1.9 GPa and 2.3 GPa around 21.5 T in experiments" (second paragraph of page 11) either in Fig 1 or Fig. 3.

It is indeed a typo, and we are grateful to the Reviewer for pointing it out. The anomalies at 1.9 GP and 2.3 GPa we refer to actually lie just under 20 T, not around 21.5 T. In Fig 3, these are the two data points (black right triangles) lie near the boundary between the $1/3$ supersolid and 10×2 supersolid. We have corrected the error and double checked the entire manuscript for consistency. In page 12, the sentence is now revised as follows:

"Interestingly, the anomalies at 1.9 GPa and 2.3 GPa just under 20 T in experiments..."

3. The observation of descendant relation among the full plaquette, 10×2 supersolid, and $1/5$ plateau phases is very interesting. Then I would expect the transition directly from 10×2 supersolid to $1/5$ plateau. However, the iPEPS result shows a wide intervening region of $1/3$ supersolid phase in between. Is there any simple explanation for this?

We thank the Reviewer for the comments. Indeed, the $1/5$ plateau and the 10×2 supersolid phases are closely related, since they are both based on the same full plaquette phase (FPP) stripe pattern. However, the 10×2 supersolid phase is not obtained by just a small deformation of the $1/5$ plateau (as it is the case for the $1/3$ supersolid phase starting from the $1/3$ plateau). The 10×2 supersolid phase is obtained starting from the $1/5$ plateau by rotating the spins on each even (odd) stripe by 90 (-90) degrees, with an additional small rotation such that there is a net magnetization in the z -direction. Thus, since the deformation is not small, we do not necessarily expect that the two phases are adjacent.

In response to the Reviewer's comment, we have added a sentence at the end of the section "*Nature of the $1/5$ plateau and 10×2 supersolid*" to clarify this point.

"Finally, the 10×2 supersolid phase can be obtained from the $1/5$ plateau by an alternating rotation of the magnetization of successive stripes clockwise or counterclockwise by 90 degrees, and by adding some magnetic particles between the stripes, see Supplementary Fig. 9. We note that, because the 10×2 supersolid is not obtained by a just small rotation of the spins of the $1/5$ plateau, we a priori do not expect it to be adjacent to the $1/5$ plateau phase."

4. Related to the previous question, since the full praquette phase does not appear in the SS model, the lowest field phase obtained by iPEPS above 1.9 GPa in Fig. 3 must be the empty praquette phase. Then It seems to me logically inappropriate to apply the above argument of descendant relation to interpret the results in Fig. 3.

We appreciate the Reviewer's following up on this point. The full plaquette phase (FPP) is energetically very close to the empty plaquette phase (EPP), and it can be stabilized as the ground state in a slightly deformed Shastry-Sutherland model, please see Ref. [24]. Thus, there is a strong competition between the two states at zero field ($H = 0$), and a close competition can also be expected at finite H . What is now very interesting is that at sufficiently large H we do not find EPP-like states, but rather FPP-like states (i.e. the new $1/5$ plateau and the 10×2 supersolid phase), already in the standard Shastry-Sutherland model without any deformation. Thus, the finite field helps to stabilize FPP-like states over the EPP-like states. In the deformed model (favoring FPP correlations), these FPP-like states naturally remain energetically favored. Furthermore, they naturally appear as descendants of the FPP, which rapidly becomes the intermediate phase upon deforming the model. We have verified this in additional calculations for the deformed model and added the results in the Supplementary Fig. 12.

Reviewer #2

The authors present a study of the material $\text{SrCu}_2(\text{BO}_3)_2$ using TDO (tunnel Diode Oscillator) technique at high pressures, high magnetic fields and low temperatures. The experimental study is accompanied by a numerical analysis based on iPEPS calculations. By combining their experimental and numerical results the authors propose a field- pressure phase diagram for $\text{SrCu}_2(\text{BO}_3)_2$ exhibiting many different magnetic phases, some of which were previously observed in that compound while others are novel.

Experiments at high pressure and furthermore, experiments combining high pressures, high magnetic fields and low temperatures are notoriously difficult. With TDO measurements, the authors present a phase diagram that covers a large range, with pressures from 0 to about 2.5 GPa and magnetic fields from 0 to 45 T. It also contains relatively precise boundaries between the different proposed phases. As such, the study is of particular interest for quantum magnetism as it provides novel information about an important compound and its theoretical counterpart, the Shastry-Sutherland model. In the TDO measurements some of the features are easy to spot while some others require more analysis and are harder to see. The later depend on consistency checks that the authors have made and that cannot be easily checked by peer review. However, in my opinion, the authors have provided reasonably sufficient justification about the location of the different features mentioned in the manuscript. The details of iPEPS calculation are not part of my expertise and I cannot comment on those. The references given are appropriate. I would therefore in principle recommend it for publication in Nature communications provided the issues raised below are answered.

We thank the Reviewer for the overall positive assessment of our work and for recommending its publication in *Nature Communications* after the issues raised are addressed.

Main concerns and questions

1. The numerical calculations make use of the pressure dependence of J and J' . However as mentioned in the text these are not well known. The ab-initio paper in Ref 26, itself contains several values and references. The $J(p)$ and $J'(p)$ chosen by the authors seem not to be any of the previously published ones. $J(p)$ is not explicitly given and $J'(p)$ is taken to vary by 3% from 0 to 1.8 GPa. First, can the slope of $J(p)$ be given. Second, this raises a question about how robust the numerical analysis is with respect to the "choice" of the pressure dependence of the exchange parameters. The authors should address this issue and explain or show how changes in the initial values of J and J' or in their slope would affect their results. In particular does the "match" between calculations and experiment in figure 3 strongly depend on those choices?

We thank the Reviewer for pointing out the confusion in our discussion. As a starting point at ambient pressure, we have taken the value of $J'/J = 0.63$ from Ref. 10 which provides the best fit to the high-field magnetization plateaus. All previous studies so far have predicted that the change in J' as a function of pressure is considerably weaker than in J . Here we have chosen $J'(p)$ to decrease linearly by 3% between zero and the critical pressure $P_c \sim 1.8$ GPa (corresponding to $J'/J=0.675$) – which lies in between the prediction based on ESR data [22] ($\sim 1\%$) and fits of exact diagonalization results of 20-site clusters to the

measured magnetic susceptibility [21] (~5%). With a 3% change in $J'(p)$, the resulting linear change in $J(p)$ is ~9.5% between ambient and critical pressure, and the corresponding slopes are ~ -0.86 K / GPa and ~ -3.08 K / GPa, respectively. In the revised version we have added this extra information about the modeling of the pressure dependence in the iPEPS results section to clarify it further. In order to see the dependence on the choice of $J'(p)$, we have added two additional phase diagrams (please see Supplementary Fig. 11), one with 1% change and the other one with 5% change, both leading to only small shifts compared to the case of 3% used for the phase diagram in the main text.

In response to the Reviewer's comment, we have made the following changes:

(1) In page 10, we expanded the discussion, and the new text now reads as follows.

"Here we model the pressure dependence assuming a linear dependence of J and J' on pressure and a small change of 3% in J' between its value at ambient pressure and its value at the critical pressure $p_c = 1.8$ GPa. This choice lies in between the prediction based on ESR data [23] (~ 1%) and magnetic susceptibility data [22] (~ 5%). At ambient pressure we use $J = 81.5$ K. This value lies in between previously predicted values [11, 44] and yields good agreement with the onset of the 1/4 and 1/3 plateaus observed in experiments. The resulting slopes of the linear functions $J'(p)$ and $J(p)$ are -0.86 K/GPa and -3.08 K/GPa, respectively."

(2) We added a new Figure as Supplementary Fig. 11 to show Phase diagrams obtained for different pressure dependence of $J'(p)$.

2. Comparison neutron vs TDO: In Figure 2, the TDO measurements are systematically higher (by 0.25 to 0.5 meV) than the spin gaps reported by neutrons and specific heat measurements in the dimer phase. In the plaquette phase however there is no shift for the low energy modes while the shift remains for the higher energy mode. Could the authors address and discuss this issue?

Is that a due to a difference in dispersion? Is that a systematic of the TDO measurement compared to other techniques. Why does the systematically higher TDO values suddenly match the neutron and specific heat values for the lower modes of the plaquette?

We thank the Reviewer for pointing out the confusion. We note that the TDO data points are associated with the right axis (external magnetic field μ_0H , in the unit of Tesla), while the spin gap reported by neutron scattering and heat capacity are plotted with the left axis (in the unit of meV). Since the two sets of data have different units, one could not directly compare them.

Moreover, the sub-1/8 anomaly occurs after the spin gap is closed and does not really represent the energy scale of the spin gap, therefore, we do not necessarily expect a close resemblance between the pressure dependence of the sub-1/8 anomaly and that of the spin gap. That is why we believe it is more appropriate to plot our TDO data in terms of the characteristic fields rather than the spin gaps, so that we do not mislead the readers. On the other hand, the low energy mode represents the transition between the plaquette phase and the AFM phase. The energy scale for such a transition could be detected by the neutron scattering and heat capacity. It is, therefore, perhaps not surprising that the pressure dependence of our TDO data follows a similar trend as that reported by the other two techniques.

In response to the Reviewer's comment, we have made the following changes:

(1) In Fig. 2, we exchanged the left and right axes, so that our TDO data is now plotted with the left axis, and the reported spin gaps are plotted with the right axis, to further clarify that our TDO data is associated with the characteristic fields not the spin gaps. The caption of Fig. 2 is revised correspondingly.

(2) We have revised the corresponding text in page 9, also, in response to the Reviewer's comment. It now reads:

"Interestingly, when plotted in the same figure, as shown in Fig. 2, the pressure-dependence of some of the characteristic fields [$\mu_0H_1(P)$ and $\mu_0H_0(P)$] and that of the spin gap [$\Delta(P)$] measured by neutron scattering and heat capacity measurements, show similar behaviors. On the other hand, some notable differences of the two types of pressure dependence are also observed at $P > 2.3$ GPa. Here, μ_0H_0 splits, signaling the emergence of the AFM state. Our observations provide a broader perspective for the evolution of the spin gap with pressure in this material."

Finally, we note that while the introduction of Mg doping does not qualitatively change the behavior of $\mu_0H_1(P)$ and $\mu_0H_0(P)$, the new modes presaging the AFM state in $\mu_0H_0(P)$ are shifted to lower energy compared to that in pure $\text{SrCu}_2(\text{BO}_3)_2$, though the doping dependence of this softening remains to be explored (see Supplementary Fig. S5 and S6 and Supplementary Note 1) ...”

Minor concern

3. a) In page 7: “H1 is the sub-1/8 anomaly that signals the onset of the condensation of triplet bound states”. Is that claim based on the current analysis and can some justification be given, or is it based on a previous work in which case a reference should be given? There is not much further discussion in the main manuscript, but in the supplementary material, it appears that it is a new interpretation. Can some more quantitative justification be provided?

We thank the Reviewer for the question. In fact, a similar question is also raised by the Reviewer 1. We kindly ask the Reviewer to please refer to our extended reply to the first comment by Reviewer 1 and please review our revised manuscript for the corresponding detailed changes.

b) Would other such “sub-plateau” anomalies be expected for the other plateaus?

A short answer would be no. We do not expect such “sub-plateau” anomaly for other plateaus. As explained also previously in our reply to the Reviewer 1, the sub-1/8 anomaly is most likely associated with the condensation of the S=2 bound states, which is a prerequisite for all other plateaus associated with crystallization of the bound states. It is expected theoretically, but its experimental evidence has not been intentionally searched for, although some experimental signatures have already been reported in Ref. 5 [Onizuka, K., et al. J. Phys. Soc. Jpn. 69, 1016 (2000)]. More detailed explanation of the sub-1/8 anomaly is now given in the revised text, as mentioned in our reply to the first comment by Reviewer 1.

4. Doped material. a) Do the indicated percentage correspond to the input composition during crystal growth or to the final result obtained. b) Were X-rays performed to verify the doping levels and the fact that the Mg impurities did actually go into the Cu positions? Nothing is said about sample characterization in the methods section.

We thank the Reviewer for the question. Our study actually builds upon some of our earlier studies using the same samples [Shi et al. Nat. Commun. 10, 2439 (2019), Ref 40]. The detailed characterization of these samples has been reported in Ref. 40. To answer the Reviewer’s question, a) yes, the indicated percentage correspond to the input composition during the crystal growth (nominal doping percentage), and b) we have conducted susceptibility measurements and extracted the Mg-impurity percentage using Curie-Weiss law. The extracted doping level agrees with the nominal value very well (please see methods and supplementary Fig. 1 in Ref 40 for details).

5. In Supplementary figures 5 and 6. There is no big shift in the dip (and bump) position from $x=0$ to $x=0.03$ doping. At $x=0.05$ there is a sudden jump. The first dip, for instance, stays close to 6.0 GPa all the way to $x=0.03$ and then jumps to ~ 4.8 GPa for $x=0.05$. The authors should discuss this peculiar behavior.

We thank the Reviewer for the question. In fact, we realize that this comment is related to the comment #2 and #8 by the Reviewer regarding the discussion of the low energy mode and its doping dependence.

Indeed, we are also intrigued by the softening behavior of the low energy mode from ~ 6 T to 4.8 T for when the doping is increased to $x=0.05$. We would like to note that the data for $x=0.02$, 0.03 plotted in Supplementary Fig. 6 were actually taken at a lower pressure (2.1 GPa) compared to that (2.4 GPa) for $x=0$, 0.05 plotted in Supplementary Fig. 5. Because of technical challenges reaching high pressure using piston-type cells, we do not have 2.4 GPa data for the $x=0.02$ and $x=0.03$ samples. We note that the splitting of the low energy mode is not observed until P is higher than 2.3 GPa. Since our data for $x=0.02$ and $x=0.03$ samples are taken at P up to 2.1 GPa, we do not know if such softening of the low energy mode at 2.4 GPa is a sudden jump or not with doping.

We have made some comments about this behavior in the previous version of our manuscript (the last sentence above section “*iPEPS calculation results*”, where it says: “*However, the impurity-driven shift in the low-field mode noted above suggests that the dopants act to destabilize the plaquette phase and instead favor the AFM phase.*”

In response to the Reviewer’s comment, we made the following changes:

(1) We further clarify this point with the following added text in page 9:

“...though the doping dependence of this softening remains to be explored (see Supplementary Figs. S5 and S6, and Supplementary Note 1).”

(2) We added the following discussion at the end of the Supplementary Note 1:

“We note that for $x=0.02$ and $x=0.03$, we do not have data up to 2.4 GPa (Supplementary Fig. 6), so the doping evolution of the softening of the low energy mode requires future exploration.”

6. a) Two orientations were measured: with H parallel to c and H perpendicular to c (i.e in the a-b plane). In the second case was the orientation systematically the same (for example H // to a) or did it change from one run to the next?

b) Should one expect a difference in magnetization or in df/dH when, for instance, H//a compared to when H is in the a-b plane at 45 deg between a and b?

We thank the Reviewer for the comment. For H parallel to the a-b plane, we have oriented the sample using Laue diffraction before our measurements, and the field is applied parallel to the a-axis (or equivalently, b-axis) within ~5 degrees. The ~ 5 degrees error comes from experimental difficulty in fixing the sample position inside the TDO coil and from mounting the TDO coil on sample stage by hand.

Moreover, the Reviewer raised an important question regarding the in-plane anisotropy of SCBO. We have indeed carefully conducted a two-axis rotation TDO experiment, where we systematically studied the in-plane anisotropy of the system. And our results reveal essentially no change in any of the characteristic field for the plateaus, suggesting same g-factors within the ab-plane (no in-plane anisotropy).

In response to the Reviewer’s comment, we have made the following changes in the “*Tunnel diode oscillator (TDO)*” section in Methods:

“The samples are oriented using Laue diffraction so that the long edge of the sample is along the a-axis (or equivalently, b-axis)... When mounting on the sample stage, the coil is oriented by hand such that the field is applied parallel to the a-axis (or b-axis) within ~5 degrees. The coil with the sample inside and a capacitor is used to form a LC circuit.”

Clarification needed

7. Not clear in text in page 6:

“similar anomalies”. Does this refer to H6 and H7 only or all of H1 to H7? “this field range”, does it refer to the field for H6 and H7 only?

We thank the Reviewer for pointing out the confusion. In both places, we refer to H6 and H7 that was defined at $P = 0$ only. Prompted by the Reviewer’s comment, we have made the following changes for clarification.

(1) In page 6, we started a new paragraph where we discussed the evolution of the two anomalies at H6 and H7 with increasing pressure.

(2) We rewrote several sentences that describe the pressure dependence of these two anomalies, and it now reads:

“We first focus on the two anomalies at the highest fields at $P = 0$, namely H6 and H7, which can be identified immediately as the onset of the 1/4 and 1/3 magnetization plateaus, respectively [40]. The natural next step is to follow the two anomalies to higher pressures. At $P = 1.1$ GPa, two similar anomalies are also observed, though shifted to lower fields (~35 T and ~40 T, respectively). In the intermediate plaquette phase, at 1.9 GPa and 2.3 GPa, we still can identify two anomalies in this field range, although they are now much weaker and shifted slightly to even lower fields. It is tempting to assign these two anomalies that we see at these high pressures (1.1 GPa, 1.9 GPa and 2.3 GPa) as extensions of the H6 (1/4 plateau) and H7 (1/3 plateau) seen at $P = 0$. However, we caution that the fate of the magnetization plateaus at higher pressure needs to be understood first.”

8. Page 8 “We first focus on H1, ...” the structure of the presentation is not very clear. One would expect then a focus on the other features, but that is not how the text continues.

We thank the Reviewer for prompting us to realize the confusing part in our presentation. Since this comment is related to the Reviewer’s comment #2, which also concerns the discussion of Fig. 2, we refer the Reviewer to please see our response above for detailed changes. These changes indeed significantly improved our presentation, and we are very grateful for the Reviewer for these comments.

9. In page 11-12, when discussing the discrepancies between calculation and experiment, the authors invoke the 3D and DM couplings and then possible structural differences related to open vs full plaquettes. All of these are acceptable as possible sources of discrepancies. However, it is not clear at the end of the paragraph why those should be important “particularly at low fields”.

What we had in mind here is that in particular the transition into the AF phase will be pushed down to lower fields by including an inter-plane coupling, since it is known that the extent of the plaquette phase gets reduced with respect to the AF phase with increasing inter-plane coupling. Also, in case the ground state in the material is an FPP ground state (and not an EPP ground state), this may also have an effect on the critical field into the AF state. At larger fields, which includes large extended phases (in contrast to the narrow AF phase in between the plaquette phase and 10x2 supersolid phase), there is a weaker competition (i.e. larger energy differences) between different phases, and thus we expect a smaller effect of the additional modifications of the model. In order to clarify further, we have extended this sentence in page 13.

“These modifications of the model may also affect the magnetization process, particularly at low fields, where the narrow AF phase is energetically closely competing with the plaquette and the 10x2 supersolid phase. We stress, however, that the 1/5 plateau and the 10x2 supersolid phase remain relevant ground states also in the deformed model (see Supplementary Fig. 12). In fact, they tend to be further stabilized by the deformation, a logical tendency since they correspond to descendants of the full plaquette state, as we discuss in the following sections.”

10. It would be good for the reader, in my opinion, to 1) explain in the text why some of the phases correspond to a “supersolid” state. For instance, what characteristics of a supersolid do these particular spin arrangements display? And 2) justify or explain the naming used such as: “10x2” and “1/3” supersolids.

We agree with the Reviewer that it would be useful to explain this. We have added additional information in the iPEPS results section to explain the meaning of the supersolid phase and the naming.

(1) On page 11, the revised text now reads:

“At high fields (up to 45 T) the dominant phases are the 1/4 plateau, the 1/3 plateau, and a 1/3 supersolid phase [11]. A supersolid phase simultaneously breaks the translational symmetry and the U(1) symmetry associated with the total Sz conservation. The 1/3 supersolid exhibits the same translational symmetry breaking pattern as the 1/3 plateau state, but with additional spin components in the transverse direction, reflecting the broken U(1) symmetry.”

(2) On page 12, the revised text now reads:

“Above the empty plaquette (P) phase in zero and small fields, we find a narrow partially polarized antiferromagnetic phase (AFM), and a 10x2 supersolid state (obtained in a 10x2 unit cell; hence the name), followed by the 1/3 supersolid and 1/3 plateau phases.”

11. Supplementary Fig S4. The quality of the data at 0.3K is much better than the data at 0.44K. Is that uniquely due to the physics or was the 0.44K data (and higher T) taken with lower statistics? It would be good to give this information in the supplementary text.

We thank the Reviewer for the acute observation which prompted us to re-examine the data shown in Supplementary Fig. 4. This figure contains three subpanels, labeled as (a), (b), and (c), and the Reviewer’s question regards the data at 1.9 GPa shown in panel (b). Here, each of the df/dH vs H trace between 0.44 K and 0.95 K is actually an average of two measurements (one is with field up-sweep and the other is with field down-sweep). For 0.3 K, we repeated the measurements for six times (three up-sweeps and three down-sweeps) and calculated the average. By repeating each measurement multiple times and following the temperature dependence, we are able to confirm the reproducibility of the identified anomalies.

Prompted by the Reviewer's comment, we noticed that when plotting the results, the data for 0.3 K trace was scaled by an additional factor of 2 compared to the rest of the temperatures. This has led to the impression that the quality of the data at 0.3 K is better than that at 0.44 K and higher temperatures. We have corrected this in the revised version of the Supplementary Fig. 4. In the corrected figure, one can see that the temperature evolution of the df/dH is gradual without sudden changes. We note, however, this does not affect any of our conclusions. We are grateful to the Reviewer for pointing this out. We have also carefully checked all other figures and made sure they are all correct.

12. Supplementary Fig 5-6. Is the vertical scale the same for all panels?

The vertical scale in Supplementary Fig. 5 and 6 are in arbitrary units, and thus different. This is because in TDO measurements, we measure the "change" in frequency of the TDO circuit. Additionally, the actual frequency of the TDO circuit can be different for each setup, however, the relative change of frequency with regards to magnetic field (thus the characteristic field at which we identify the anomalies) should be reproducible – this has indeed been the case for the presented results.

Reviewer #3

The manuscript by Shi et al combines experimental and theoretical efforts to explore the magnetic phase diagram of $\text{SrCu}_2(\text{BO}_3)_2$ under extreme conditions: sub-Kelvin temperatures, high pressures and high field. The anomalies observed in the tunnel diode oscillator (TDO) experiment are then used to establish the phase boundaries. By comparing to the iPEPS simulations, the authors claim the discoveries of a few new phases, in particular the 10×2 supersolid and a $1/5$ plateau.

While it is always interesting to find novel magnetic states, for example the 10×2 supersolid and the $1/5$ plateau, I am not fully convinced that this work meets the criteria of validity and novelty to publish under Nature Communications. My major concerns are list below:

1. As the authors admitted themselves (last paragraph in "iPEPS calculation results"), the model they are using does not even correctly reproduce the "full"-plaquette ground state at high pressure and zero field. Consequently, the theoretical findings at high pressure are sitting on a wrong base. This naturally explains their discrepancy in matching the theoretical and experimental data at pressures larger than 1.9 GPa. Consequently, I am not convinced that the 10×2 supersolid phase and the narrow AF phase are relevant to the real material.

We are grateful to the Reviewer for careful reading of our manuscript, and we fully understand the importance to address the criticism raised.

We would like to begin with stressing on the fact that the Shastry-Sutherland model (SSM) has been an excellent starting point to understand several physical phenomena in SCBO over the past 20 years. At ambient pressure it reproduces the sequence of magnetization plateaus observed in experiments [10,12], and also thermodynamic quantities are in excellent agreement [43]. At finite pressure, it reproduces the transition into a gapped plaquette phase, followed by a transition into an antiferromagnetic phase, and it correctly reproduces a critical point at finite temperature [26], in agreement with experiments. Thus, there is clear evidence from many previous studies, that the model is relevant for SCBO, and for this reason, we have used this model also as a starting point in the present study.

Now, there exist indications from experiments that the intermediate plaquette phase is not the empty plaquette phase (EPP), but a full plaquette phase (FPP) – whether this is really the case is still an open question. However, in any case, these two phases lie energetically very close. In Ref. [24] it was shown that already a small deformation of the standard SSM stabilizes the FPP phase over the EPP phase. Since the deformed model lies very close to the original model in parameter space, we a priori do not expect substantial changes of the phase diagram at high magnetic fields upon slightly modifying the model.

Another concern raised by the Reviewer may be that if there is a strong competition between EPP and FPP phases for $h=0$, the same may be true at finite h , which would imply a strong sensitivity of the results upon deforming the model. However, this is not the case. In fact, the new phases we find (i.e. the new $1/5$ plateau and the 10×2 supersolid) are FPP-like states, and not EPP-like states. These FPP-like states will naturally remain the ground state also of the deformed model (which favors FPP correlations). To demonstrate this numerically we have performed additional simulations for the deformed model, for two different types of

deformations, which clearly show that the $1/5$ plateau and the 10×2 supersolid states remain the ground state as a function of deformation, see new Supplementary Fig. 12.

Thus, to summarize, we believe at present the standard Shastry-Sutherland model is still the best and most natural starting point to investigate the physics of SCBO at high pressure and high fields, and further evidence is provided by the remarkable agreement between numerical simulations and experiment measurements we find for the high-field plateaus over a wide range of pressures. Even if SCBO would be better described by a deformed model (whose effective coupling parameters are still unclear at present), the new $1/5$ plateau and 10×2 supersolid phases we find here, will definitely also be relevant for the deformed model with an FPP ground state, as we demonstrated in the additional numerical simulations.

In response to the Reviewer's comment, we have made the following changes:

(1) At the end of the iPEPS section, we added the following sentences:

"We stress, however, that the $1/5$ plateau and the 10×2 supersolid phase remain relevant ground states also in the deformed model (see Supplementary Fig. 12). In fact, they tend to be further stabilized by the deformation, a logical tendency since they correspond to descendants of the full plaquette state, as we discuss in the following sections"

(2) We added Supplementary Fig. 12.

2. In Fig. 3, There is only one experimental data point near the upper phase boundary of the 10×2 supersolid phase, and there is no matching experimental/numerical data point at the lower phase boundary. This again questions the validity of the 10×2 phase.

In fact, the lower phase boundary of the 10×2 phase from experiments is given by the black stars and pentagons in the phase diagram in Fig. 3. Like the transition between plaquette and AF phase, this phase boundary is found at lower fields in experiments than in the numerical simulations. As discussed in our manuscript, this deviation could stem from the additional interlayer coupling, which is known to reduce the extent of the plaquette phase.

3. For the $1/5$ plateau and $1/3$ supersolid, there are only limited experimental data points for the phase boundaries (three~four for $1/5$ plateau and two~three for $1/3$ supersolid). The interpretation of these phase are mostly relying on the model which could not even reproduce the zero phase full plaquette phase, and there is no direct experimental evidence.

We hope that the above explanation has convinced the Reviewer that the Shastry-Sutherland model is relevant for SCBO in this range of field and pressure, since all the conclusions drawn for this model regarding the high field phase diagram would still apply if a small distortion was included to stabilize the full plaquette phase in zero field. Besides, we would like to emphasize that there is no adjustable parameter in the calculation – to clarify this point further, please note that we did not adjust any parameters to fit the phase diagram; in fact, we used a plausible dependence of J and J' on pressure, obtained based on the previous work, and derived the phase diagram on the basis of this choice without making any further adjustments. So, the fact that the experimental points, even if there are only a few of them, are consistent with the boundaries of the theoretical parameter-free phase diagram in this field and pressure range, should in our view be taken as a strong argument in favor of our interpretation.

4. Even if the numerically discovered new phases are established, I am not convinced yet that these results are of high significance to the field that qualifies publication in a high profile journal like the Nature Communication. Besides showing novel magnetic structures, the authors should provide evidence that such findings have significance either on the conceptual or at the application level.

We believe there are at least two good reasons to publish this manuscript in *Nature Communications*: (1) it demonstrates the unique power of TDO to probe materials in combined extreme conditions of field and pressure; (2) it predicts the stabilization in a field of new types of plateaus beyond those already identified (Wigner crystals of triplets and Wigner crystals of bound states of triplets). These phases can be interpreted as Mott insulating phases of hard-core bosons, the role of the magnetic field being played by the chemical potential. This unexpected finding might thus have implications in other contexts where interacting bosonic models are relevant.

To further clarify the significance of our results, we have added the following sentences at the end of the “Discussion”:

“Moreover, our results not only demonstrate the unique power of TDO in probing materials in combined extreme conditions of field and pressure, but also predict the stabilization of new types of plateaus beyond those already identified (Wigner crystals of triplets and Wigner crystals of bound states of triplets). These phases can be interpreted as Mott insulating phases of hard-core bosons, the role of the magnetic field being played by the chemical potential. This unexpected finding might thus have implications in other contexts where interacting bosonic models are relevant.”

We are grateful to all three Reviewers again for their careful reading of our manuscript, and for the valuable comments, which have prompted us to improve our manuscript significantly. As demonstrated in this reply and in our revised manuscript, we have made extensive changes to make sure we have fully addressed all the Reviewers’ concerns. Therefore, we hope that the Reviewers finds our response satisfactory and our revised manuscript suitable for publication in *Nature Communications*.

REVIEWER COMMENTS

Reviewer #1 (Remarks to the Author):

The authors have adequately responded to my questions and comments in their reply to review reports, from which I was assured that the manuscript should have been revised appropriately to be ready for publication. However, I found that the revisions promised by the authors in the reply were not included in the revised manuscript. I suspect that the authors may have resubmitted a wrong version not the final one by mistake.

Following is the list of inconsistencies between the reply to the reviewers #1 and the revised manuscript.

Reply to the comment 2: It is indeed a typo, and we are grateful to the Reviewer for pointing it out. The anomalies at 1.9 GPa and 2.3 GPa we refer to actually lie just under 20 T, not around 21.5 T. ... In page 12, the sentence is now revised as follows: "Interestingly, the anomalies at 1.9 GPa and 2.3 GPa just under 20 T in experiments..."

However, the revised manuscript still reads "Interestingly, the anomalies at 1.9 GPa and 2.3 GPa around 21.5 T in experiments..."

Reply to the comment 3: In response to the Reviewer's comment, we have added a sentence at the end of the section "Nature of

the $1/5$ plateau and 10×2 supersolid" to clarify this point.

"Finally, ... see Supplementary Fig. 9. We note that, because the 10×2 supersolid is not obtained by a just small rotation of the spins of the $1/5$ plateau, we a priori do not expect it to be adjacent to the $1/5$ plateau phase."

However, the last sentence is not included in the revised manuscript.

I would like just to confirm that proper revisions are made in the final version.

Reviewer #2 (Remarks to the Author):

Following the numbering of the previous correspondence:

1) I am quite confused by the percentages referring to the pressure dependence of the coupling $J(p)$ and $J'(p)$ given and used by the authors

1A) It appears to me that the extraction of the percentages given by the authors namely 5% from magnetic susceptibility and 1% from ESR are erroneous. (May be due to a reading from the left axis in the corresponding figures instead of the axis on the right (?)).

In the reference for magnetic susceptibility (Zayed et al 2014 figure 2b) reading from the figure's right axis I get:

$$J'(p) = 47\text{K} - 4.7 \text{ K/GPa} * p, \text{ from extracting: } J'(0)=47 \text{ K and } J'(3 \text{ GPa})=33 \text{ K}$$

$$J(p) = 76\text{K} - 10 \text{ K/GPa} * p, \text{ from extracting: } J(0)=76 \text{ K and } J(3 \text{ GPa})=46 \text{ K}$$

This give a decrease of 18% for J' from 0 to 1.8Gpa. Namely $J'(0)=47\text{K}$ and $J'(1.8\text{Gpa})= 38.6\text{K}$

And a decrease of 24% for J from 0 to 1.8Gpa. Namely $J(0)=76\text{K}$ and $J'(1.8\text{Gpa})= 58\text{K}$

In the ESR paper (Sakurai et al 2018) the equations are given:

$$J(p) = 69.1\text{K} - 5.14 \text{ K/GPa} * p ; \text{ thus: } J(0)=69.1 \text{ K and } J(1.8 \text{ GPa})=59.8 \text{ K}$$

$$\alpha(p) = 0.601 + 0.0322 * p \text{ [they linearize alpha not } J']$$

$$J'(p) = J(p) * \alpha(p), \text{ so that}$$

$$J'(0)=41.5\text{K and } J'(1.8)=39.4 \text{ K; which to me is a change of 5% not 1% from 0 to 1.8 GPa.}$$

1B) I am also confused by the values given by the authors in their reply.

If I understand correctly, they use

$J(0)=81.5\text{K}$ and $J'(0)=51.35\text{K}$ from the mention that $(J'/J=0.63$ at ambient pressure).

The slopes are given as -3.08 K/GPa for J and -0.86K/GPa for J'

This gives

$$J'(p)=51.35\text{K} - 0.86\text{K/GPa} * p$$

$$J(p)=81.5\text{K} - 3.08\text{K/GPa} * p$$

$$J'(1.8\text{GPa})=49.8\text{K} \text{ (indeed 3\% smaller than } J'(0)=51.345\text{K)}$$

$$J(1.8\text{GPa})=76.0\text{ K} \text{ which is 6.8 \% smaller than } J(0)=81.5\text{ K} \text{ not 9.5\%}.$$

And J'/J at 1.8 GPa is 0.655 not 0.675 indicated in the reply.

Could the author clarify this (in particular is $J'(0)=51.35\text{K}$?, or maybe it is that they linearize α and not J ?) , and clearly give the two linear functions used.

1C) Based on the values extracted above, the authors cannot justify the use of 3% as being between the ESR and susceptibility values since those are 5% and 18% respectively. The authors should readdress this issue and provide a new justification for the couplings they used or use different values.

1D) Finally since this section needs to be readdressed, could the authors also address the change in initial values ($J(0)$ and $J'(p)$) as they have addressed the change of slope in the supplementary figure 11.

The author have addressed the concerns for 2), 3) and 6) to 12).

I give some additional comments about those:

3) and the main manuscript, it refers to a fig 1.c when there is none.

6) The text in the methods now makes it clearer that all the measurements in this study are with H/a and that the H/c , where sub 1/8 is not seen, is a reference to a previous article.

12) Here the idea behind the question was, when comparing the different concentrations, if some additional info could be extracted from the relative depths (and widths) of the dips not only their positions, therefore the vertical scale would have mattered.

Points 4) and 5)

Surprisingly in Ref 40 methods, no indication is given about the characterization of $x=0.05$ sample. It only mentions the $x=0.02$ and 0.03 samples. As the author indicated in their response, the field was different for $x=0.05$, which should be the main reason for the “sudden” softening. Is it also possible to rule out the unlikely hypothesis that this could be related to some properties of the $x=0.05$, as it is not very clear how it was characterized?

General comment

The authors have improved the manuscript and taken into account the remarks by the reviewers. The issues raised in point 1 and possibly the issue regarding the characterization of sample $x=0.05$ should be addressed.

If the issue regarding point 1 can be suitably resolved I would recommend publication.

Reviewer #3 (Remarks to the Author):

Through extra calculations with the deformed model which captures the full-plaquette phase (Supplementary Fig. 12), the authors have convinced me that the $1/5$ plateau and the 10×2 supersolid phase are indeed relevant phases for the $\text{SrCu}_2(\text{BO}_3)_2$ compound. While the authors did not provide enough experimental data points for the newly discovered phases, in my view the application of the state-of-the-art TDO technique already demonstrates great potential in probing materials in combined extreme conditions.

Therefore, I am now in a positive position in recommending the publication of this work.

Reply to the Reviewers

We are grateful to all three Reviewers for carefully reading our reply and our revised manuscript and are pleased that they all find our response satisfactory except for some minor issues, which we have fully addressed in our latest version of the manuscript. The valuable comments of the Reviewers have helped us significantly improve the manuscript and we hope that the Reviewers now find it suitable for publication. The changes are summarized below (comments in blue, replies to comments in black):

Reviewer #1

The authors have adequately responded to my questions and comments in their reply to review reports, from which I was assured that the manuscript should have been revised appropriately to be ready for publication. However, I found that the revisions promised by the authors in the reply were not included in the revised manuscript. I suspect that the authors may have resubmitted a wrong version not the final one by mistake.

Following is the list of inconsistencies between the reply to the reviewers #1 and the revised manuscript.

We thank the Reviewer for pointing out the inconsistencies between our revised manuscript and our reply. We inadvertently resubmitted a version of the manuscript that did not include some of the corrections mentioned in our response letter, and we are deeply sorry for that. We have carefully checked our revised manuscript and verified that all of the corrections are now included.

Reply to the comment 2: It is indeed a typo, and we are grateful to the Reviewer for pointing it out. The anomalies at 1.9 GP and 2.3 GPa we refer to actually lie just under 20 T, not around 21.5 T. ... In page 12, the sentence is now revised as follows: "Interestingly, the anomalies at 1.9 GPa and 2.3 GPa just under 20 T in experiments..."

However, the revised manuscript still reads "Interestingly, the anomalies at 1.9 GPa and 2.3 GPa around 21.5 T in experiments..."

We have now addressed this issue: on page 12, we have replaced "...around 21.5 T..." with "...just under 20 T..."

Reply to the comment 3: In response to the Reviewer's comment, we have added a sentence at the end of the section "Nature of the 1/5 plateau and 10x2 supersolid" to clarify this point.

"Finally, ... see Supplementary Fig. 9. We note that, because the 10x2 supersolid is not obtained by a just small rotation of the spins of the 1/5 plateau, we a priori do not expect it to be adjacent to the 1/5 plateau phase."

However, the last sentence is not included in the revised manuscript.

The sentence that was missing ("We note that...1/5 plateau phase.") was included in our earlier version as a footnote (Footnote #2) at the bottom of the page. We apologize for not having stated this more clearly in the response letter.

Prompted by the Reviewer's comment, we realize that the footnote is not used in *Nature Communications* per its guidelines. In this revised version, we have moved the sentence in the footnote to the main text where it is cited.

Apart from the above points, we can confirm that all revisions were included in the revised manuscript.

I would like just to confirm that proper revisions are made in the final version.

We have gone through the manuscript again and confirmed that all of the revisions have been applied. We hope the Reviewer finds our revised manuscript now suitable for publication in *Nature Communications*.

Reviewer #2

We thank the Reviewer for their very careful reading of the revised manuscript, and for spotting errors accidentally introduced during the first revision.

Following the numbering of the previous correspondence:

1) I am quite confused by the percentages referring to the pressure dependence of the coupling $J(p)$ and $J'(p)$ given and used by the authors

1A) It appears to me that the extraction of the percentages given by the authors namely 5% from magnetic susceptibility and 1% from ESR are erroneous. (May be due to a reading from the left axis in the corresponding figures instead of the axis on the right (?)).

In the reference for magnetic susceptibility (Zayed et al 2014 figure 2b) reading from the figure's right axis I get:

$$J'(p) = 47K - 4.7 \text{ K/GPa} * p, \text{ from extracting: } J'(0)=47 \text{ K and } J'(3 \text{ Gpa})=33 \text{ K}$$

$$J(p) = 76K - 10 \text{ K/GPa} * p, \text{ from extracting: } J(0)=76 \text{ K and } J(3 \text{ Gpa})=46 \text{ K}$$

This give a decrease of 18% for J' from 0 to 1.8Gpa. Namely $J'(0)=47K$ and $J'(1.8Gpa)= 38.6K$

And a decrease of 24% for J from 0 to 1.8Gpa. Namely $J(0)=76K$ and $J'(1.8Gpa)= 58K$

In the ESR paper (Sakurai et al 2018) the equations are given:

$$J(p) = 69.1K - 5.14 \text{ K/GPa} * p ; \text{ thus: } J(0)=69.1 \text{ K and } J(1.8 \text{ GPa})=59.8 \text{ K}$$

$$\alpha(p) = 0.601 + 0.0322 * p \text{ [they linearize alpha not } J']$$

$$J'(p) = J(p) * \alpha(p), \text{ so that}$$

$$J'(0)=41.5K \text{ and } J'(1.8)=39.4 \text{ K; which to me is a change of 5% not 1% from 0 to 1.8 GPa.}$$

We are grateful to the Reviewer for this important observation. This issue was indeed due to coarse extraction of the data from those papers. The accurate values are indeed ~5% from ESR and ~17-18% from magnetic susceptibility. We have corrected the values in the paper accordingly. As detailed below, we also have included a discussion and new data (Fig. 3, Supplementary Figs. 11 and 12), which demonstrate qualitatively the same phase diagram that we have established. The iPEPS calculation results with the updated parameters are shown to be in good agreement with our TDO experimental results.

1B) I am also confused by the values given by the authors in their reply.

If I understand correctly, they use

$$J(0)=81.5K \text{ and } J'(0)=51.35K \text{ from the mention that } (J'/J=0.63 \text{ at ambient pressure}).$$

The slopes are given as -3.08 K/Gpa for J and -0.86K/Gpa for J'

This gives

$$J'(p) = 51.35K - 0.86K/GPa * p$$

$$J(p) = 81.5K - 3.08K/GPa * p$$

$$J'(1.8GPa) = 49.8K \text{ (indeed 3% smaller than } J'(0)=51.345K \text{)}$$

$$J(1.8GPa) = 76.0 \text{ K which is 6.8 \% smaller than } J(0)=81.5 \text{ K not 9.5\%}.$$

And J'/J at 1.8 GPa is 0.655 not 0.675 indicated in the reply.

Could the author clarify this (in particular is $J'(0)=51.35K$?, or maybe it is that they linearize alpha and not J ?), and clearly give the two linear functions used.

We apologize for the confusion, and we thank the Reviewer for pointing this out.

The slope -3.08K/GPa for $J(p)$ should be changed to -4.29K/GPa when considering the 3% change, i.e., the two linear functions are:

$$J'(p) = 51.35\text{K} - 0.86\text{K/GPa} \cdot p$$

$$J(p) = 81.5\text{K} - 4.29\text{K/GPa} \cdot p$$

So J'/J should be 0.675 at 1.8 GPa.

In the newly added Supplementary Fig. 12, we also show the robustness of our results against choices of slightly different J'/J .

1C) Based on the values extracted above, the authors cannot justify the use of 3% as being between the ESR and susceptibility values since those are 5% and 18% respectively. The authors should readdress this issue and provide a new justification for the couplings they used or use different values.

We fully agree with the Reviewer. To address this issue, in the revised version we plot the phase diagram in Fig. 3 using the 5% value from ESR, which still provides very good agreement between experiment and theory. We note that the ESR result is also close to the estimate (4%) found in a recent ab-initio study (Ref. 26), and we believe it is more reliable than the much higher value (17%) obtained from the indirect method of using fits to magnetic susceptibility data.

In page 10 of the manuscript, we have changed the text accordingly:

“Applying pressure leads to an effective increase of the ratio J'/J ; however, the precise pressure dependence of J' and J is not known. Here we model the pressure dependence by linear functions for $J(p)$ and $J'(p)$, with a change of 5% in J' between its value at ambient pressure and its value at the critical pressure $p_c = 1.8$ GPa [corresponding to $J'/J = 0.675$ (Ref.36)]. A change of 5% is also predicted from ESR data²³, and is close to the estimate (4%) obtained in a recent ab-initio study²⁶ (in contrast, a substantially larger value (17%) was found from fits to magnetic susceptibility data²²). At ambient pressure, we use $J = 81.5$ K, which lies in between previously predicted values^{11, 44} and yields good agreement with the onset of the 1/4 and 1/3 plateaus observed in experiments. The resulting slopes of the linear functions $J'(p)$ and $J(p)$ are -1.43 K / GPa and -5.13 K / GPa, respectively. In Supplementary Figs. 11 and 12 we present alternative phase diagrams using different parameter sets in order to illustrate the dependence of the phase boundaries on the various parameters. The boundaries change marginally, as long as the variation of J' is only a few percent. The change by 17% deduced from fitting the susceptibility²² would by contrast lead to a phase diagram whose boundaries depart significantly from our experimental data and thus can be discarded.”

1D) Finally since this section needs to be readdressed, could the authors also address the change in initial values ($J(0)$ and $J'(p)$) as they have addressed the change of slope in the supplementary figure 11.

Prompted by the Reviewer's suggestion, we have now added 8 alternative phase diagrams to the supplemental material (Supplementary Figs. 11 and 12), illustrating the change when varying the percentage of the change in $J'(p)$, the value of $J(0)$, the value of $J'(0)/J(0)$, and the value of $J'(p_c)/J(p_c)$. As shown in these figures, our results are robust against small changes in these parameters, and the agreement between iPEPS calculations and TDO results are generally good.

The author have addressed the concerns for 2), 3) and 6) to 12).

I give some additional comments about those:

3) and the main manuscript, it refers to a fig 1.c when there is none.

We thank the Reviewer for catching our typo. In this case, “Fig. 1c” should actually be “Fig. 1b”. We have made the corrections on pages 6 and 7.

6) The text in the methods now makes it clearer that all the measurements in this study are with $H//a$ and that the $H//c$, where sub 1/8 is not seen, is a reference to a previous article.

12) Here the idea behind the question was, when comparing the different concentrations, if some additional info could be extracted from the relative depths (and widths) of the dips not only their positions, therefore the vertical scale would have mattered.

We are glad that the Reviewer finds our response to the point 6) and 12) satisfactory. We appreciate the Reviewer’s consideration with regards to question 12. Indeed, it could help evaluate the effect of doping quantitatively if the relative scale of the magnetization changes could be compared among different doping concentrations. However, we note that we aim to understand the doping effect at high pressure and high magnetic field qualitatively in this current study, and a quantitative investigation of the doping effect will be the subject of a future study.

Points 4) and 5)

Surprisingly in Ref 40 methods, no indication is given about the characterization of $x=0.05$ sample. It only mentions the $x=0.02$ and 0.03 samples. As the author indicated in their response, the field was different for $x=0.05$, which should be the main reason for the “sudden” softening. Is it also possible to rule out the unlikely hypothesis that this could be related to some properties of the $x=0.05$, as it is not very clear how it was characterized?

We thank the Reviewer for the question, which has prompted us to further clarify the sample information as discussed below.

As the Reviewer points out, the highest pressure that we could achieve for the $x=0.05$ and $x=0$ samples was 2.4 GPa – whereas the highest pressure accessed for $x=0.02$ and 0.03 samples was 2.1 GPa. This is also noted at the end of our Supplementary Note 1. In our earlier study (Ref. 40), we characterized the Mg doping concentration for $x=0.02$ and 0.03 using Curie-Weiss fits and found that the experimentally determined values match well with the nominal doping levels. Moreover, though we did not conduct a Curie-Weiss fit analysis for $x=0.05$ in Ref. 40, we did, however, have systematic doping-dependence data for magnetization, magnetostriction, and TDO (e.g. Figs. 1 and 3 in Ref. 40), all of which show a consistent, gradual evolution of the magnetization behavior with increasing doping. Therefore, we believe the doping concentration for $x=0.05$ should also be close to its nominal value.

That being said, we believe the softening of the low energy mode (onset of the AFM state) above ~ 2.3 GPa mostly likely is gradual with increasing doping. This is because (1) the effect of doping on the magnetization process is gradual at $P=0$ (Ref. 40), and (2) the AFM state is favored at higher T , H , and P where entropy is increased (see Ref. 27). It is not surprising that increased impurity level by Mg doping also destabilizes the plaquette phase and favors the AFM phase. The detailed mechanism of doping effects on the AFM state, however, is not the focus of the paper and is beyond the scope of our current study.

The following changes are made in the revised manuscript:

(1) In the “Methods” section, we have added:

“...In Ref. 40, the experimentally extracted doping concentrations using Curie-Weiss fits were found to match well with the nominal doping levels for $x=0.02$, 0.03 , and a gradual evolution of the magnetization behavior with increasing doping up to $x=0.05$ was observed.”

(2) In the caption of Supplementary Fig. 6, we have added the following sentence:

“...Note that the pressure (2.1 GPa) is lower than the one (2.4 GPa) used in Supplementary Fig. 5, so caution is needed when making a comparison.”

(3) In page 9, we have revised the text in the second paragraph, it now reads:

“...(see Supplementary Fig. 5 for $x=0, 0.05$ data collected at 2.4 GPa, Supplementary Fig. 6 for $x=0.02, 0.03$ data collected at 2.1 GPa, and Supplementary Note 1)...”

(4) In page 10, just above the section “iPEPS calculation results”, we added:

“...which is perhaps not surprising as we will show below that the AFM phase is also favored at higher T , H , and P .”

(5) At the end of the Supplementary Note 1, we have revised and expanded the last sentence, so it now reads:

“...We note that for the $x=0.02$ and $x=0.03$ samples, we have data only up to 2.1 GPa (Supplementary Fig. 6), where the AFM phase is still negligible (Fig. 3). Based on our observations at ambient pressure², it is more likely that the softening of the low energy mode (onset of the AFM phase) is a gradual change with doping. The detailed doping evolution of the softening of the low energy mode will require further exploration.”

General comment

The authors have improved the manuscript and taken into account the remarks by the reviewers. The issues raised in point 1 and possibly the issue regarding the characterization of sample $x=0.05$ should be addressed.

If the issue regarding point 1 can be suitably resolved I would recommend publication.

We thank the Reviewer again for the detailed and helpful comments. As shown above, we have made further changes to address the Reviewer’s remaining questions about points 1, 4 and 5. We hope the Reviewer finds our response satisfactory and our revised manuscript suitable for publication in *Nature Communications*.

Reviewer #3

Through extra calculations with the deformed model which captures the full-plaquette phase (Supplementary Fig. 12), the authors have convinced me that the $1/5$ plateau and the 10×2 supersolid phase are indeed relevant phases for the $\text{SrCu}_2(\text{BO}_3)_2$ compound. While the authors did not provide enough experimental data points for the newly discovered phases, in my view the application of the state-of-the-art TDO technique already demonstrates great potential in probing materials in combined extreme conditions.

Therefore, I am now in a positive position in recommending the publication of this work.

We appreciate the Reviewer’s positive assessment of our revised manuscript and for recommending its publication in *Nature Communications*.

We agree with the Reviewer that the state-of-the-art TDO technique indeed demonstrates great potential in probing materials in combined extreme conditions. We hope that our study will motivate future studies in

exploring previously inaccessible phase spaces using the TDO technique, not only in $\text{SrCu}_2(\text{BO}_3)_2$, but also in other strongly-correlated materials.

We again express our gratitude to all three Reviewers for their constructive comments that have helped us improve our manuscript significantly. As demonstrated in this reply and in our revised manuscript, we have now fully addressed all the Reviewers' remaining concerns. We hope that our revised manuscript is now suitable for publication in *Nature Communications*.

REVIEWERS' COMMENTS

Reviewer #2 (Remarks to the Author):

The authors have addressed the concerns raised in the last review and I recommend the corrected manuscript for publication.

Regarding the presentation of the results, the captions of Supplementary Figures 11 and 12 contain a very dense amount of information on the different choices of parameters. I would suggest for the benefit of the readers, to add a table containing all the linear functions used (or some alternative presentation that would increase the readability). As this suggestion is straightforward it should not necessitate further review.

Reply to the Reviewers

We are grateful to Reviewer #2 again for carefully reading our revised manuscript and recommending publication after a minor change, which we have fully addressed. The changes are summarized below (comments in blue, replies to comments in black):

Reviewer #2

The authors have addressed the concerns raised in the last review and I recommend the corrected manuscript for publication.

Regarding the presentation of the results, the captions of Supplementary Figures 11 and 12 contain a very dense amount of information on the different choices of parameters. I would suggest for the benefit of the readers, to add a table containing all the linear functions used (or some alternative presentation that would increase the readability). As this suggestion is straightforward it should not necessitate further review.

We thank the Reviewer for suggestions that further helps improve the readability of our paper. In response to the Reviewer's comment, we have made the following changes in the Supplementary Material.

(1) We have added Supplementary Table 1 which summarizes all the linear functions used in Fig. 3 and Supplementary Figs. 11 and 12, as suggested by the Reviewer #2.

(2) We have rewritten and streamlined the captions of Supplementary Figs. 11 and 12 to further improve readability.

In the page 10 of main text, we have referred to the newly added Supplementary Table 1, and the revised sentence reads:

"In Supplementary Figs. 11 and 12 and Supplementary Table 1 we present alternative phase diagrams using different parameter sets, ..."